# Aerosol effects on deep convection: The propagation of aerosol perturbations through convective cloud microphysics

Max Heikenfeld[1], Bethan White[1,2], Laurent Labbouz[1,3], and Philip Stier[1]

[1]Atmospheric, Oceanic and Planetary Physics, Department of Physics, University of Oxford, Oxford, United Kingdom
[2]ARC Centre of Excellence for Climate System Science, School of Earth, Atmosphere and Environment, Monash University, Melbourne, Australia
[3]Laboratoire d'Aérologie, Université de Toulouse, CNRS, UPS, Toulouse, France

**Correspondence:** Max Heikenfeld (max.heikenfeld@physics.ox.ac.uk)

**Abstract.** The impact of aerosols on ice- and mixed-phase processes in deep convective clouds remains highly uncertain, and the wide range of interacting microphysical processes are still poorly understood. To understand these processes, we analyse diagnostic output of all individual microphysical process rates for two bulk microphysics schemes in the Weather and Research Forecasting model (WRF). We investigate the response of individual processes to changes in aerosol conditions and the propagation of perturbations through the microphysics all the way to the macrophysical development of the convective clouds. We perform simulations for two different cases of idealised supercells using two double-moment bulk microphysics schemes and a bin microphysics scheme. The simulations cover a comprehensive range of values for cloud droplet number concentration (CDNC) and cloud condensation nuclei (CCN) concentration as a proxy for aerosol effects on convective clouds. We have developed a new cloud tracking algorithm to analyse the morphology and time evolution of individually tracked convective cells in the simulations and their response to the aerosol perturbations.

This analysis confirms an expected decrease in warm rain formation processes due to autoconversion and accretion for more polluted conditions. There is no evidence of a significant increase in the total amount of latent heat, as changes to the individual components of the integrated latent heating in the cloud compensate each other. The latent heating from freezing and riming processes is shifted to a higher altitude in the cloud, but there is no significant change to the integrated latent heat from freezing. Different choices in the treatment of deposition and sublimation processes between the microphysics schemes lead to strong differences including feedbacks onto condensation and evaporation. These changes in the microphysical processes explain some of the response in cloud mass and the altitude of the cloud centre of gravity. However, there remain some contrasts in the development of the bulk cloud parameters between the microphysics schemes and the two simulated cases.

## 1 Introduction

Deep convective clouds are an important feature of the Earth's atmosphere, ranging from widespread convection dominating the atmosphere in the tropics to mid-latitude convective systems (Emanuel, 1994). The impact of aerosols on ice- and mixed-phase processes in convective clouds remains highly uncertain (Tao et al., 2012; Varble, 2018), which has implications for determining the role of aerosol-cloud interactions in the climate system. Representing these effects in global climate models

poses additional challenges due to the relatively small length scales often less than a few kilometres at which convective clouds develop and because of limitations in the representations of microphysical processes in the convective parametrisations (Tao et al., 2012; Boucher et al., 2013; Sullivan et al., 2016) with only few models explicitly representing the effects of aerosols on deep convective clouds (e.g. Song and Zhang, 2011; Guo et al., 2015; Kipling et al., 2017; Zhang et al., 2017; Labbouz et al.,

2018). The highly localised nature of convective processes also leads to major challenges in observations both from satellites and aircraft measurements (Rosenfeld et al., 2014).

Over recent years numerous studies using cloud-resolving model simulations (CRM) have investigated aerosol-convection interactions in various setups, ranging from case study simulations to idealised simulations of squall lines or supercells like the cases used in this study (Seifert and Beheng, 2006; Storer et al., 2010; Morrison, 2012; Kalina et al., 2014). The results,

however, vary strongly between many of these studies. The differences can be attributed to the simulation of different types of convection, different environmental conditions like humidity or wind shear, but are also related to differences between the models or modelling approaches used (Tao et al., 2012; Fan et al., 2016; White et al., 2017). These challenges in modelling are strongly related to numerous interacting physical processes (Fan et al., 2016) in cloud microphysics and to the interaction between clouds and other processes in the atmosphere on different scales (Tao et al., 2012). In addition to the analysis of process

rates in numerical simulations, analytical evaluations of the microphysical rate equations of the microphysics schemes can give important insights into the propagation of aerosol effects in the cloud microphysics (Glassmeier and Lohmann, 2016). This kind of analytical approach works well for warm-phase clouds but is less conclusive for the response of mixed-phase clouds, especially deep convective clouds, due to many compensating effects and the complexity of the processes involving ice-phase hydrometeors (Glassmeier and Lohmann, 2016).

Convective invigoration (Andreae et al., 2004; Rosenfeld et al., 2008; Lebo and Seinfeld, 2011) has been proposed as a mechanism by which aerosols impact the development of deep convective clouds. A higher number concentration of aerosols suitable to act as cloud condensation nuclei (CCN) can lead to more but smaller cloud droplets, which are less likely to be processed into rain and precipitated out of the cloud. This would lead to more water reaching the freezing level in the cloud where subsequent freezing leads to additional latent heating in the higher levels of the cloud, enhancing the strength of the convection with

higher updraft speeds and cloud top height. Other studies point out the additional impact of the larger number of aerosols, and subsequently cloud droplets, leading to smaller ice particles which then favours increased cloud fraction, cloud top height, and cloud thickness (Fan et al., 2013) due to reduced fall speeds of the ice particles. This implies a significant radiative effect on the climate system through enhanced anvils (Koren et al., 2010). Grabowski and Morrison (2016) argue that the effects can be purely attributed to the effects of smaller droplets and ice crystals with negligible effects of the thermodynamic enhancement

proposed in Rosenfeld et al. (2008). Some of the differences in the assessments of convective invigoration due to aerosols are actually caused by the difference in the definition of both changes in aerosol and the quantification of the strength of convection based on different variables such as surface precipitation, updraft speeds or cloud top heights (Lebo et al., 2012; Altaratz et al., 2014). Significant mechanisms buffering the impact of aerosols on clouds and precipitation, both with a focus on warm-phase processes (Stevens and Feingold, 2009) and for mixed-phase and ice-clouds (Fan et al., 2016) have been proposed. However,

recent studies question the attribution of observed relationships between aerosol concentrations and cloud-top height to aerosol

microphysical effects (Varble, 2018; Nishant and Sherwood, 2017). It is, therefore, one of the main goals of this paper to investigate if and how these proposed mechanisms of convective invigoration, especially the proposed invigoration of convection due to additional latent heat release from freezing, manifest themselves in numerical simulations.

Many studies have pointed out the representation of cloud microphysics in models as one of the main sources of uncertainty in high-resolution model studies of aerosol-cloud interactions or cloud feedbacks to a warming climate, especially for mixed-phase and ice-phase clouds (Tao et al., 2012; Khain et al., 2015; White et al., 2017). This also holds for the role of the microphysics schemes in global model simulations of both convection and aerosol-cloud interactions (Lohmann and Feichter, 2005; Gettelman, 2015; Malavelle et al., 2017).

Most currently used cloud microphysics schemes can be separated into two approaches, bulk microphysics schemes and bin microphysics schemes (Khain et al., 2015). Bulk microphysics schemes assume a specific size distribution for a range of different hydrometeor classes and describe their evolution and interactions based on a certain number of moments of these distributions. Double moment schemes with both prognostic mass and number concentrations of the hydrometeors are the current standard and necessary to meaningfully represent aerosol-cloud interactions (Khain et al., 2015; Igel et al., 2014).

The separation of the hydrometeors into individual hydrometeor classes in microphysics schemes brings with it specific challenges in resolving the microphysical processes. In bulk schemes, liquid water in the cloud is separated into cloud droplets and raindrops. The collision-coalescence processes leading to the formation of rain from cloud droplets have to be parametrised through the artificial process of droplet autoconversion and a simplified treatment of accretion of droplets by raindrops. The semi-empirical nature of these parametrisations has been shown to be the source of major uncertainty in the assessment of aerosol-cloud interactions in numerical model simulations (Khain et al., 2015; White et al., 2017). In the ice phase, most current microphysics schemes separate the hydrometeors into a number of different classes such as pristine ice, snow, hail or graupel. The equations and parameters for the calculation of the microphysical process rates as well as important physical properties of the hydrometeors, such as shape, density or the specific form of the size distribution are specified for each individual hydrometeor class. These choices additionally impact important physical processes such as the fall speeds of hydrometeors in the calculation of sedimentation or the radiative properties of the hydrometeors. This can lead to abrupt changes to the evolution of the cloud due to a change in the partition between the hydrometeor classes in the ice phase of the cloud (Morrison and Milbrandt, 2014). There have been developments towards overcoming the separation of ice hydrometeors into fixed individual classes (Harrington et al., 2013a, b; Morrison and Milbrandt, 2014; Morrison et al., 2015) by treating ice-phase hydrometeors as one single class with smoothly varying physical properties, which have been implemented both in cloud-resolving models and in global climate models. Nevertheless, most current applications rely on microphysics schemes performing the separation into different hydrometeor classes. Better understanding the possible effects and causes of shifts in the hydrometeor partitions through the comprehensive analysis of the microphysical pathways in the two bulk microphysics schemes is thus a main focus of this paper.

Bin microphysics schemes represent the different hydrometeors in the cloud through a number of individual size bins per hydrometeor class, thus allowing for more flexible representation of the actual size distribution and the interaction between the different size bins (Khain et al., 2015). Due to the large number of simulated variables, however, this approach results in high

computational cost. One of the main benefits is avoiding the artificial separation between cloud droplets and raindrops that causes challenges in bulk microphysics scheme for example in the form of a parametrisation of the autoconversion processes (Khain et al., 2015). The representation of ice-phase hydrometeors in typical bin microphysics schemes, however, is based on separate hydrometeor classes as in the bulk schemes, each individually resolving their size distribution through a number of

bins (Khain et al., 2015). While many studies have proposed that bin-resolving microphysics schemes are necessary to reliably represent possible microphysical aerosol effects on convective clouds (Khain et al., 2004; Fan et al., 2012, 2016) in model simulations, a large range of studies and applications, e.g. routine numerical weather prediction (NWP), coupled simulations with a complex aerosol- and chemistry and global climate model simulations as well as a large number of CRM based studies of aerosol-cloud interactions apply bulk microphysics schemes.

This study aims to unravel the underlying microphysical mechanisms responsible for the large diversity of simulated aerosol effects on convection through a comprehensive analysis of the propagation of aerosol perturbations through microphysical pathways in different microphysics schemes.

Tracking individual convective cells in the simulation makes it possible to draw direct conclusions about the behaviour of individual convective cells in the simulations, e.g. regarding their time evolution or the response to changes in simulation pa-

rameters that go beyond the bulk average over the simulation domain or the sum of all cloudy areas in the simulation. The analysis of tracked cumulus clouds has been applied in previous studies (e.g Dawe and Austin, 2012; Heus and Seifert, 2013; Heiblum et al., 2016a, b) with a focus on various aspects of convective clouds including the effects of aerosol perturbations on deep convection (Terwey and Rozoff, 2014).

We have implemented detailed microphysical process rate diagnostics for pathway analysis in the two double-moment mi-

crophysics schemes of Morrison et al. (2009) and Thompson et al. (2004). We analyse the cloud morphology and the spatial structure of the microphysical processes in individual tracked convective cells. We display the microphysical process rates in the form of scaled pie charts. This has been inspired by previous studies using this type of visualisation of the spatiotemporal development of physical processes for other applications. Schutgens and Stier (2014) performed a pathway analysis for the aerosol processes in a global climate model (ECHAM-HAM). Chang et al. (2015) applied a microphysical pathway analysis

including a similar visualisation of process rates to simulations of pyro-convective clouds, however, using a much simpler two-dimensional model for highly idealised individual clouds.

In addition to the detailed process rate diagnostics, we derive important bulk cloud properties, such as the total cloud mass or the altitude of the centre of gravity and analyse their evolution over the life cycle of the tracked cells. Our approach goes beyond previous studies with a similar setup (Morrison et al., 2009; Kalina et al., 2014) that mainly focussed on domain average

properties and only a specific subset of microphysical processes.

We use a well-documented idealised supercell setup based on Weisman and Klemp (1982, 1984), that was applied in previous studies (e.g. Khain and Lynn, 2009; Morrison et al., 2009; Kalina et al., 2014), to create a well-defined development of a strong convective cell, allowing us to focus on the microphysical evolution of individual convective cells. To test the representativeness of our results from this first case, we include simulations for a second idealised supercell case based on the measurements

and model setups from Kumjian et al. (2010); Naylor and Gilmore (2012); Dawson et al. (2013).

We represent idealised aerosol perturbations through changes to a fixed cloud droplet number concentration (CDNC) in each simulation with the two bulk microphysics schemes. This allows us to isolate the actual cloud microphysical pathways from uncertainties in the representation of the activation of cloud condensation nuclei (CCN) in numerical models (Ghan et al., 2011; Simpson et al., 2014; Rothenberg et al., 2018). Simulations are performed for a comprehensive range of CDNC for each microphysics scheme ranging from values representative of very clean, maritime conditions (CDNC=50 $cm^{-3}$) to very polluted situations (CDNC=2500 $cm^{-3}$).

We compare the results to simulations performed with a bin microphysics scheme (HUJI spectral bin scheme) for a subset of the analyses to investigate whether the effects investigated in more detail through the microphysical pathway analysis for the two bulk microphysics schemes agree with the response of a bin microphysics scheme to perturbations of aerosol proxies.

## 2 Methods

### 2.1 Model Setup

The simulations are performed with the Weather and Research Forecasting model (WRF) version 3.7.1 (Skamarock et al., 2005). We use the two-moment microphysics schemes from Thompson et al. (2004, 2008), denoted as THOM, and from Morrison et al. (2005, 2009), called MORR in our figures and tables. To isolate the role of cloud microphysics for aerosol effects on deep convection from additional uncertainties in model-simulated aerosol fields, we apply a fixed cloud droplet number concentration (CDNC) in the two bulk microphysics schemes for each simulation. In each of the schemes, the CDNC is reset to the chosen value at the end of each model time step in all cloudy grid points. We vary this CDNC value between different simulations as a proxy for aerosol number concentration. There are versions of both bulk microphysics schemes that include the activation of a fixed CCN spectrum or even interactive aerosols (Thompson and Eidhammer, 2014; Wang et al., 2013). However, the implementation of both the cloud droplet activation and the representation of the aerosol distributions is very different between the two microphysics schemes, which would add additional differences between the schemes compared to representing the perturbations in the form of a varying CDNC.

The detailed analyses of the process rates in this paper are carried out for simulations using the two bulk microphysics schemes. To investigate how the results obtained from the detailed analysis of the two bulk microphysics schemes hold for a bin cloud microphysics scheme, we also include additional simulations with the Hebrew University cloud model (HUCM) spectral-bin microphysics scheme (Khain et al., 2004; Lynn et al., 2005a, b), called SBM in the rest of the paper. We perform a subset of the analyses for this microphysical scheme, excluding the detailed microphysical process rate analysis but including the analysis of changes to the hydrometeor mixing ratios and the bulk cloud properties. We use the full version of the spectral bin microphysics scheme in WRF (Khain et al., 2012) and perform a variation of CCN number concentration.

Both bulk microphysics schemes make use of saturation adjustment, removing all water vapour exceeding the saturation vapour pressure in each time-step and instantaneously condensing it to cloud water at each time step. This prevents a build-up of supersaturation in strong updrafts and can thus impact effects of perturbations in the microphysics (Lebo et al., 2012). The bin microphysics scheme (SBM) includes an explicit calculation of supersaturation in the microphysics at each time step and

allows for a build-up of supersaturation in strong updrafts over several time steps.

We simulate two different idealised supercell cases. The first set of simulations (CASE1) is based on the default WRF quarter-circle shear supercell case representative of a supercell case over the Southern Great Plains of the United States (Khain and Lynn, 2009; Lebo and Seinfeld, 2011). This case uses an initial sounding described in Weisman and Klemp (1982) with a surface temperature of $300\,\mathrm{K}$ and a surface vapour mixing ratio of $14\,\mathrm{g\,kg^{-1}}$. The wind profile is taken from Weisman and Rotunno (2000) and features a wind shear of $40\,\mathrm{ms^{-1}}$ made up of a quarter-circle shear up to a height of $2\,\mathrm{km}$ and a linear shear further up to $7\,\mathrm{km}$ height. The initiation of convection is triggered by a warm bubble with a magnitude of $3\,\mathrm{K}$ in potential temperature centred at $1.5\,\mathrm{km}$ height in the centre of the domain with a radius of $10\,\mathrm{km}$ horizontally and $1.5\,\mathrm{km}$ vertically in which the perturbation decays with the square of the cosine towards the edge of the bubble (Morrison, 2012). This type of setup has been used for a number of similar studies in the past (Storer et al., 2010; Morrison and Milbrandt, 2010; Morrison, 2012; Kalina et al., 2014).

To test the representativeness of the results for different cases of idealised deep convection, a set of simulations for a second supercell case (CASE2) is based on an observed supercell storm over Oklahoma in 2008 (Kumjian et al., 2010). In contrast to the first case, the initial profiles are from observations used in the model experiments in Dawson et al. (2013). This case features a significantly drier initial profile with a surface temperature of $308\,\mathrm{K}$ and a surface water vapour mixing ratio of $16\,\mathrm{g\,kg^{-1}}$ along with wind shear of similar magnitude to CASE1. The initiation of convection in this case is created by forced convergence near the surface based on nudging the vertical velocity over the same volume that is used for the warm bubble in CASE1. The methodology is described in detail in Naylor and Gilmore (2012) and we use an updraft speed peaking at 5 $\mathrm{m\,s^{-1}}$ at the centre of the volume.

Both cases are simulated without a boundary layer scheme and the calculation of surface fluxes or radiation. The horizontal grid spacing of the simulations is $1\,\mathrm{km}$ to sufficiently resolve the main features of the simulated supercell. We use a model domain size of 84 grid cells in each horizontal dimension and open boundary conditions on each side of the modelling domain. The vertical resolution of the 96 model layers varies from about $50\,\mathrm{m}$ at the surface to $300\,\mathrm{m}$ at the top of the model. Simulations are performed with a time-step of 5 seconds. The standard model diagnostics and the microphysical pathway diagnostics (Section 2.3) are output every 5 minutes to sufficiently resolve the development of the microphysical processes during the life cycle of the deep convective clouds.

## 2.2 Variation of aerosol proxies: CDNC or CCN

We analyse the effects of varying the cloud droplet number concentration (CDNC) in the two bulk microphysics schemes to isolate the impact of microphysical pathways. We use a CDNC of $250\,\mathrm{cm^{-3}}$ as a baseline simulation. Simulations are performed for two CDNC values corresponding to a cleaner environment than the baseline simulation ($50\,\mathrm{cm^{-3}}$ and $100\,\mathrm{cm^{-3}}$) and five values representing more polluted conditions ($500\,\mathrm{cm^{-3}}$, $1000\,\mathrm{cm^{-3}}$, $1500\,\mathrm{cm^{-3}}$, $2000\,\mathrm{cm^{-3}}$ and $2500\,\mathrm{cm^{-3}}$).

For the simulations with the spectral-bin microphysics scheme, activation of aerosols to cloud droplets is calculated from a cloud condensation nuclei (CCN) spectrum following the equation $N_C = N_0 * S^k$, with the prognostic supersaturation $S$, the particle number concentration $N_0$ and an exponent $k$. The exponent is kept fixed at $k = 0.5$, while $N_0$ is varied in a range from

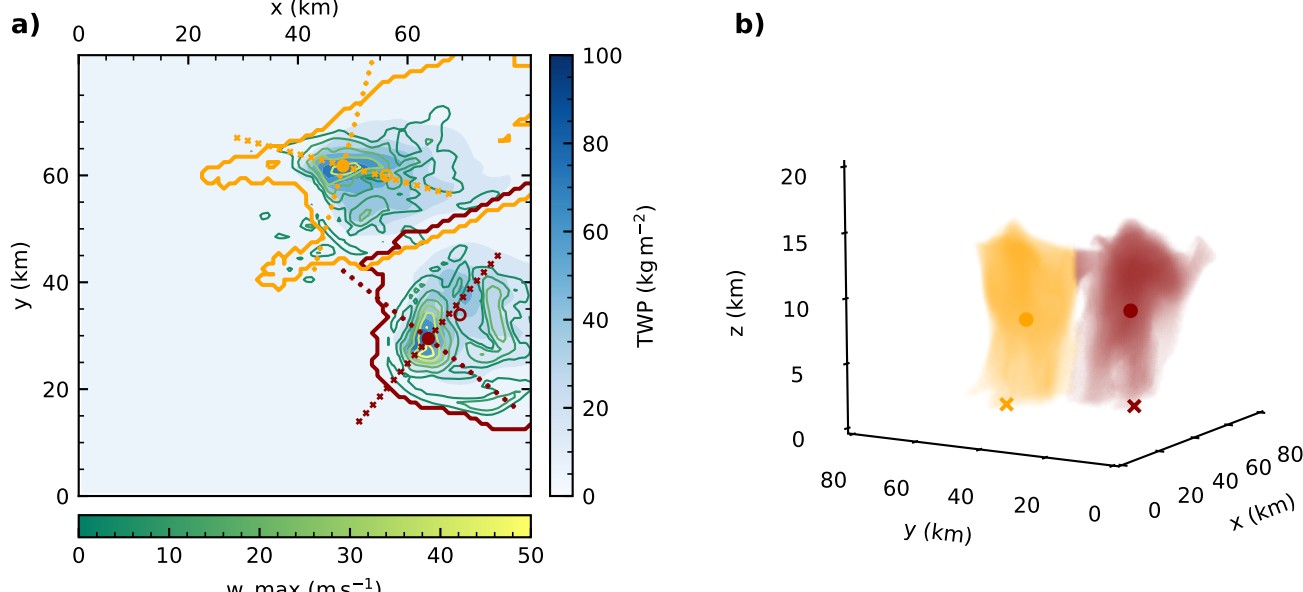

**Figure 1.** a) Illustration of the result of the tracking and watershedding methodology after 90 minutes of simulation time with the total water path field in blues and contours of column maximum vertical velocities in greens. The filled circles represent the tracked updraft cores, while the empty circles show the position of the centre of gravity determined by the watershedding algorithm. Crosses denote the slices along/across the line of travel of the cell that are used for the analysis of the cloud morphology. The coloured contour lines represent the projection of the respective cloud mask for each cell to the surface. b) 3D rendering of the $1\,\mathrm{g\,kg^{-1}}$ condensate mixing ratio threshold of the two tracked cells in the simulation at the same point in time including the horizontal location of the tracked updraft (cross) and centre of gravity (dot).

$75\,\mathrm{cm^{-3}}$ to $6750\,\mathrm{cm^{-3}}$. This yields cloud droplet number concentrations with median values spanning a similar range to those chosen for the two bulk microphysics schemes (Table 1).

## 2.3 Pathway analysis

We have extended two double-moment bulk microphysics schemes, the Morrison scheme (Morrison et al., 2005, 2009) and the
5  Thomson scheme (Thompson et al., 2004, 2008) in WRF 3.7.1, by writing detailed microphysical pathway diagnostics at each output time step. This includes all individual process rates for both hydrometeor mass and hydrometeor number mixing ratio as well as individual latent heating rates for the three phase transitions (liquid-vapour, liquid-ice, ice-vapour) and the hydrometeor mass and number tendencies for the individual hydrometeor classes (cloud water, rain, cloud ice, graupel, snow) are diagnosed at every output time step.

10 For most analyses in this study, the individual microphysical processes are grouped into a consistent set of classes according to their contribution to the hydrometeor mass transfer in the model. This includes the six different phase transitions between

**Table 1.** Overview of the 52 simulations performed in this study, including the two cases simulated and the different CDNC/CCN values for each of the microphysics schemes. The CDNC for the SBM simulations are the median values for grid points with a cloud water mixing ratio larger than $10 \, \mathrm{g \, kg^{-1}}$.

| Case | Microphysics | CDNC (cm$^{-3}$) | CCN (cm$^{-3}$) |
|---|---|---|---|
| **CASE1**<br>Weisman and Klemp (1982, 1984) | **MORR**<br>Morrison et al. (2005, 2009) | **50, 100, 250, 500, 1000,**<br>**1500, 2000, 2500** | - |
| | **THOM**<br>Thompson et al. (2004, 2008) | **50, 100, 250, 500, 1000,**<br>**1500, 2000, 2500** | - |
| | **SBM**<br>Khain et al. (2004)<br>Lynn et al. (2005a, b) | 12, 28, 54, 128, 419,<br>648, 870, 1310, 1753, 2194 | **67.5, 135, 270, 540, 1350**<br>**2025, 2700, 4050, 5400, 6750** |
| **CASE2**<br>Naylor and Gilmore (2012)<br>Dawson et al. (2013)<br>Kumjian et al. (2010) | **MORR**<br>Morrison et al. (2005, 2009) | **50, 100, 250, 500, 1000,**<br>**1500, 2000, 2500** | - |
| | **THOM**<br>Thompson et al. (2004, 2008) | **50, 100, 250, 500, 1000,**<br>**1500, 2000, 2500** | - |
| | **SBM**<br>Khain et al. (2004)<br>Lynn et al. (2005a, b) | 12, 25, 47, 171, 393<br>603, 819, 1239, 1657, 2078 | **67.5, 135, 270, 540, 1350**<br>**2025, 2700, 4050, 5400, 6750** |

frozen hydrometeors, water drops and water vapour (*condensation*, *evaporation*, *freezing* including riming, *melting*, *deposition* and *sublimation*) as well as the warm *rain formation* due to autoconversion and accretion of cloud droplets and all processes that transfer mass between the different frozen hydrometeors as *ice processes*. For some of the more detailed analyses, this grouping is performed in a more detailed way, e.g. separating freezing and riming processes or splitting them up by the specific

5   hydrometeor class involved in the transfer. A collection of all the individual microphysical process rates represented in the two bulk microphysics schemes including the grouping discussed here is given in the appendix (Table A1 for the Morrison microphysics scheme and in Table A2 for the Thompson microphysics scheme).

## 2.4   Convective cell tracking

10   We have developed a tracking algorithm focussed on the tracking of individual deep convective cells in CRM simulations, but flexible enough to be extended to other applications, e.g. simulations of shallow convection or based on geostationary satellite observations using brightness temperature data. The initial tracking of features is performed on the column maximum vertical velocity at each output time step using the python tracking library trackpy (Allan et al., 2016). These features are then filtered and linked to consistent trajectories. The trajectories are extrapolated to two additional output time steps at the start and at the

end to allow for the inclusion of both the initiation of the cell and the decaying later stages of the cell development.

Based on these trajectories, a three-dimensional watershedding algorithm *morphology.watershed* from the python image processing package scikit-image (van der Walt et al., 2014) is applied to the total condensed water content field (mass mixing ratio of all hydrometeors) at each output time step to infer the volume of the cloud associated with the tracked updraft. We use a threshold of $1\,\mathrm{g\,kg}^{-3}$ to define the core cloudy grid points in the simulations. A variation of this threshold by up to an order of magnitude to $0.1\,\mathrm{g\,m}^{-3}$ only showed minor changes to the results of the study.

A separate watershedding is performed for both liquid water content (cloud droplets and rain drops) and ice water content (all ice hydrometeors). This allows for the determination of the centre of gravity and the mass, for the entire cloud as well as for the in-cloud liquid and frozen phase, respectively. The evolution of the centre of gravity has been studied mainly for warm convective clouds (e.g. Koren et al., 2009; Dagan et al., 2015, 2017, 2018) and with a focus on the warm phase of deep convective clouds (Chen et al., 2017).

The tracking algorithm does not explicitly treat splitting and merging of convective cells. In all simulated cases in this study, the initial convective cell splits into two separate counter rotating cells early into the simulations. In CASE1 this leads to a relatively symmetric situation with similarly strong individual cells. In both cases, one of the cells develops more directly out of the initial cell, in CASE1 this is the right-moving cell, while in CASE2 this is the stronger left moving cell. In each simulation, this stronger cell gets picked up as a continuation of the initial cell by the tracking algorithm. The second cell has been analysed following the same methodology and showed very similar results in all major aspects. We have thus decided to focus on the analysis of the first cell in this paper and to not discuss the results from the second cell in more detail.

Microphysical process rates, latent heating rates and other cloud microphysical parameters such as hydrometeor mixing ratios are summed up for regularly-spaced altitude intervals in the volume of the individual cells to get representative profiles for each cloud. We interpolate the microphysical process rates and other variables used in the analysis to slices along and perpendicular to the line of travel of the cell (Fig. 1) to visualise and analyse the morphology of the cells for different simulation setups and at different stages of the cloud life cycle.

## 3 Results

### 3.1 Baseline simulations

The simulations with $\mathrm{CDNC} = 250\,\mathrm{cm}^{-3}$ for both bulk microphysics schemes (Fig. 2 and Fig. 3) are used as a baseline simulation representative of intermediate aerosol loading. As for all the following figures for CASE1, these analyses are based on a combination of the initial stage of the cell and the right-moving cell after the cell split. We use three different points in time (15 minutes, 25 minutes and 60 minutes) to illustrate the microphysical evolution of the cell in simulations with the two different microphysics schemes.

During the initial phase of the formation of the convective cloud in the simulation using the Morrison bulk microphysics scheme (Fig. 2 a,d,g), the two major microphysical processes are condensation to form cloud droplets and rain formation from these droplets, while the top of the cloud at around $7.5\,\mathrm{km}$ is already influenced by freezing and riming processes. The simu-

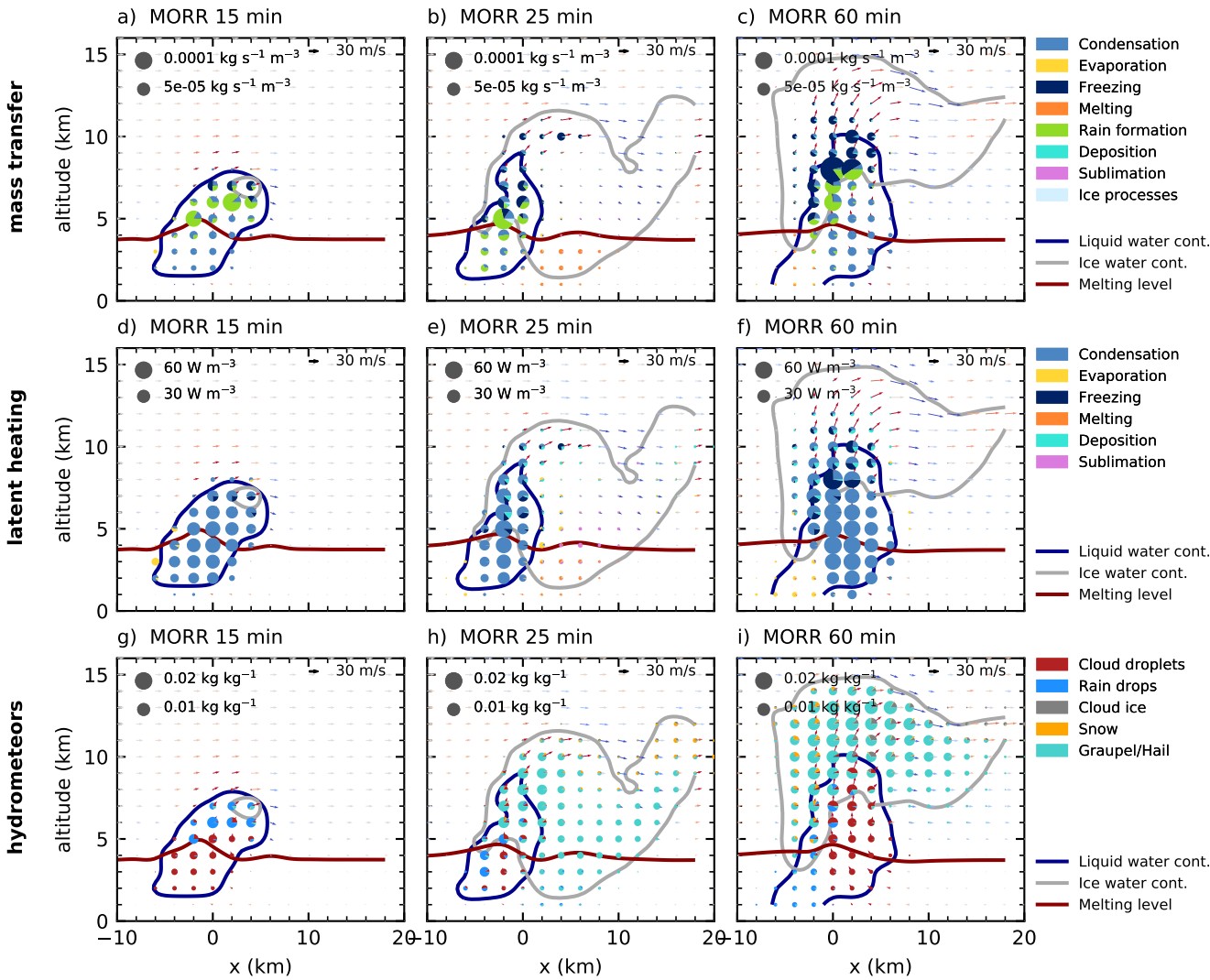

**Figure 2.** Cloud microphysical morphology along a slice parallel to the cell track for a cloud droplet number concentration of $250\,\mathrm{cm^{-3}}$ for the Morrison microphysics scheme. The area of each specific colour in the pie charts is proportional to the water turnover (a-c) in $\mathrm{kg\,m^{-3}\,s^{-3}}$ and latent heating (d-f) in $\mathrm{W\,m^{-3}}$ for the process rates and to the mass mixing ratio for the hydrometeors (g-i). Contour lines denote the mixing ratio threshold of 1 g/kg for liquid (blue) and frozen (grey) water content as well as the melting level ($0°$C isotherm). Arrows denote the wind field with updrafts in red and downdrafts in blue.

lation with the Thompson microphysics scheme shows a similar development during the initial cloud stage (Fig. 3 a,d,g). The initiation of freezing at the top of the cloud is slightly delayed in comparison to the simulation with the Morrison scheme. During the next 10 minutes, the cell quickly intensifies, dominated by the development of rain formation (autoconversion of

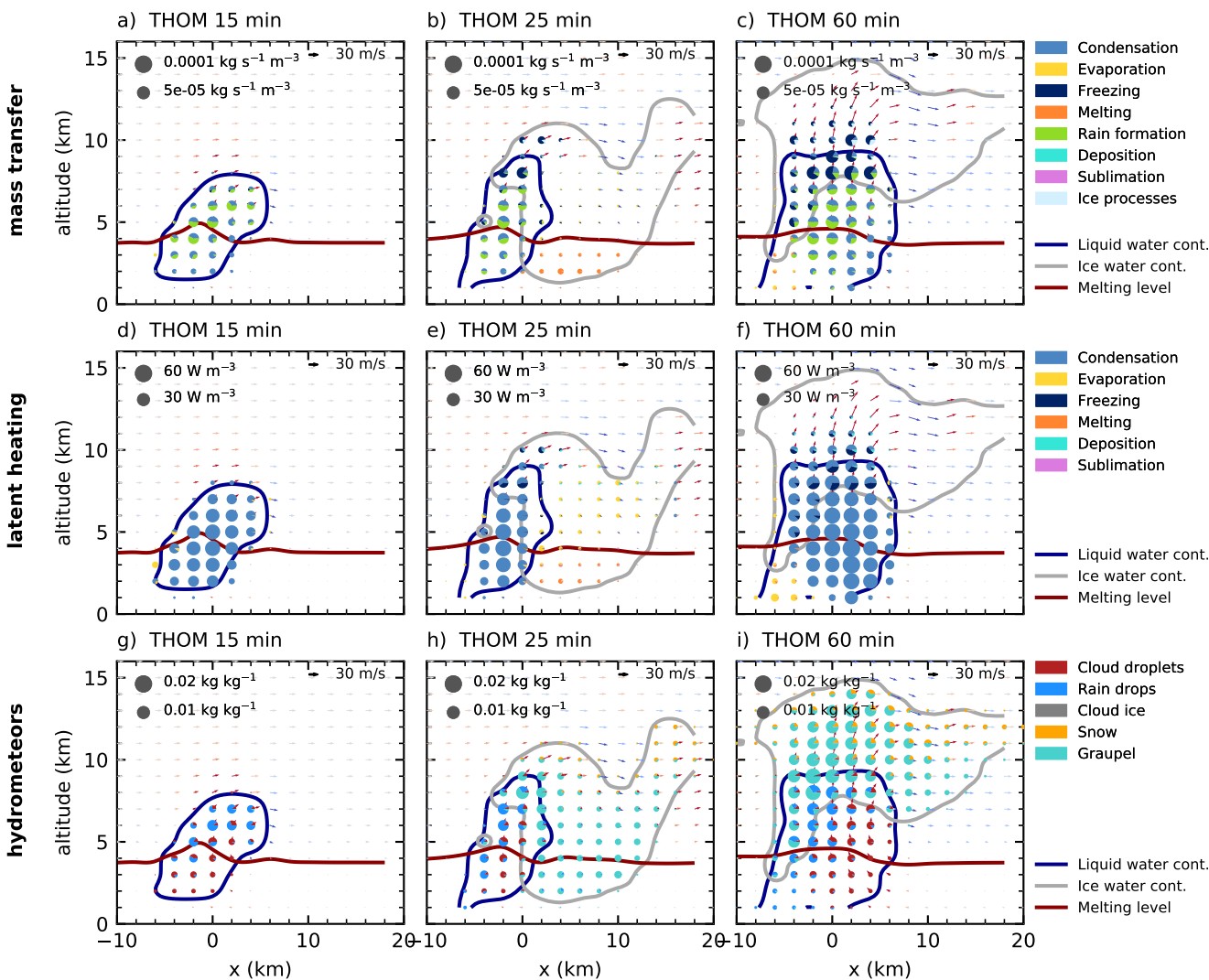

**Figure 3.** Cloud microphysical morphology along a slice parallel to the cell track for a cloud droplet number concentration of $250\,\text{cm}^{-3}$ for the Thompson microphysics scheme as in Fig. 2.

cloud droplets and accretion of cloud droplets by rain) between 4 and $7\,\text{km}$. Freezing occurs at a height of about $7\text{-}8\,\text{km}$. After an hour of simulation, the cell has developed into a mature supercell with hail dominating the mass mixing ratio in the ice phase. A significant amount of cloud droplets extends up to $10\,\text{km}$ height. Rain formation and freezing occur in the region of the strongest updraft with a width of about $5\,\text{km}$ for both microphysics schemes. During the later stage, the freezing in the simulation using the Morrison microphysics scheme takes place over a substantial vertical range and is strongest at both edges of the mixed-phase region of the cloud at around $8\,\text{km}$ and $10\,\text{km}$ altitude (Fig. 2 c). The Thompson scheme instead shows a

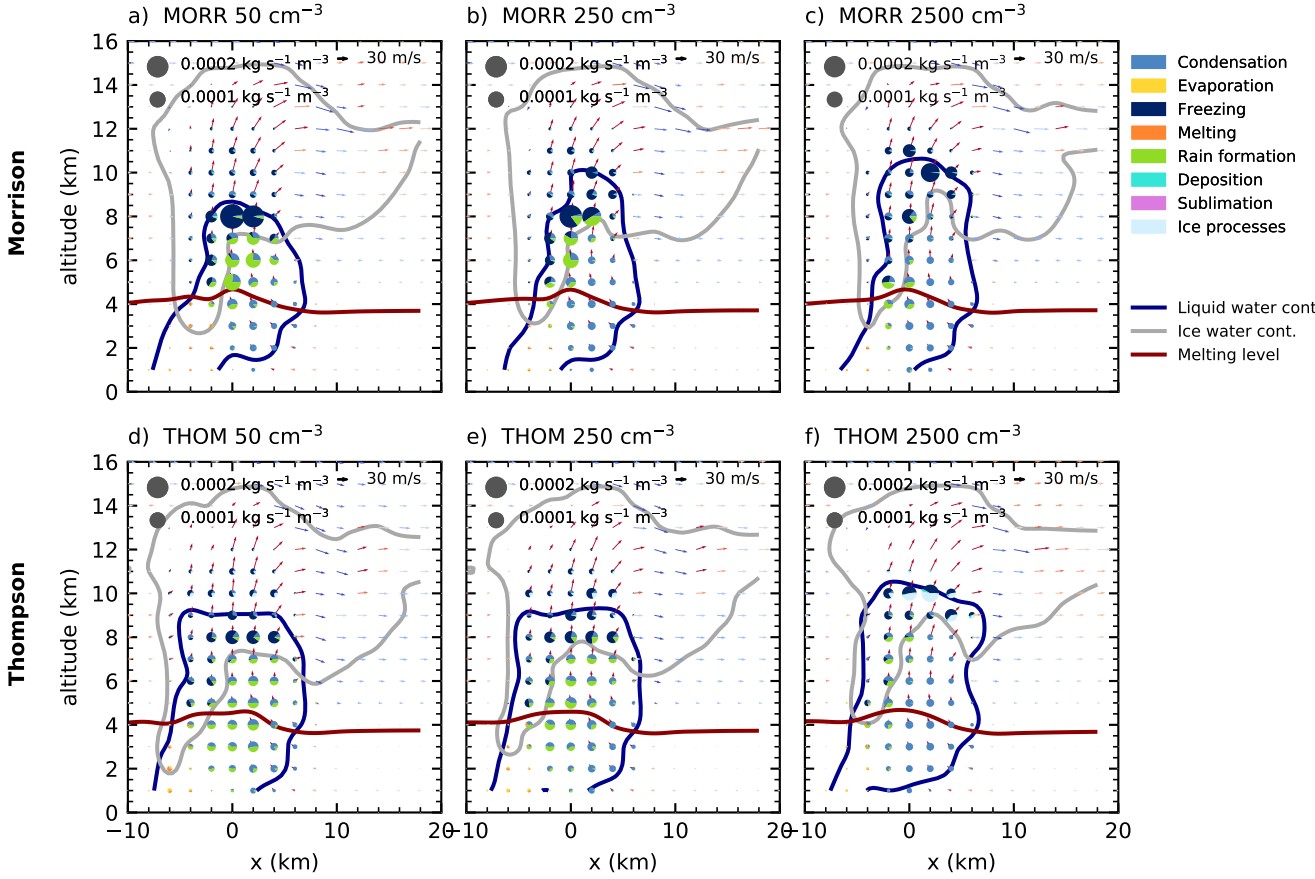

**Figure 4.** Cloud microphysical morphology along a slice through the cloud parallel to the track of the cell for simulations with three different CDNC values (left: $50\,\mathrm{cm}^{-3}$, middle: $2500\,\mathrm{cm}^{-3}$, right: $2500\,\mathrm{cm}^{-3}$) after 60 minutes of simulation using the two bulk microphysics schemes (top: Morrison, bottom: Thompson).

more confined region of freezing. In both bulk microphysics schemes, condensation processes dominate the latent heat release in the cloud for all stages of the cloud development (Fig. 2 d-f, Fig. 3 d-f). In the mature stage of the cell, the main difference in the hydrometeor classes between the two microphysics schemes is an enhanced presence of snow both in the core and in the anvil for the Thompson microphysics scheme (Fig. 2 i and Fig. 3 i).

## 3.2 Effects on cloud morphology and microphysical process rates

We first investigate changes to the right-moving cell in the CASE1 due to a variation of CDNC. We focus on three different CDNC values (clean, baseline, polluted, see Fig. 4) after 60 minutes of simulations using the two bulk microphysics schemes. In the microphysical process rates, a decrease of rain formation from droplets (autoconversion and accretion) with increasing CDNC is evident in the core of the cell for both bulk microphysics schemes. For both bulk schemes, the freezing and riming

processes are shifted upwards with increasing CDNC. The mixed phase region of the cloud, indicated by the liquid water mixing ratio contour in Fig. 4, extends about 1-2 km higher in the polluted case for each bulk scheme.

In the hydrometeor mass mixing ratios (Fig. 5), an increase in cloud droplet mass at the expense of raindrops for increasing CDNC is evident in both bulk microphysics schemes and the spectral bin microphysics scheme, particularly in the mixed phase region of the cloud at around 6-8 km). In the Thompson scheme, most of the ice-phase hydrometeor mass is present in the form of snow for the high CDNC simulation (Fig. 5 d), especially towards the cloud top and in the anvil region, while graupel dominates except in the anvil for the cleanest case (Fig. 5 c). In contrast, the ice-phase in the Morrison scheme shows a high hail mixing ratio for low and high CDNC values (Fig. 5 a,b) and additional ice particles, but only small amounts of snow in the simulation with the highest CDNC value. The simulations using the spectral bin microphysics scheme (Fig. 5 e,f) show a stronger increase in cloud droplet mass mixing ratio than the two bulk schemes for increased CCN. Graupel and hail, the predominant ice-phase hydrometeors in the cleanest simulation, get replaced by cloud ice particles for the highest CCN value. However, it has to be taken into account that the definition of the hydrometeor classes differs between the three different microphysics schemes.

Fig. 6 provides a vertically resolved view of the time evolution of the microphysical process rates over the life cycle of the right-moving cell for the two bulk microphysics schemes under the cleanest and most polluted conditions. For both schemes, a strong decrease in the warm rain formation processes (autoconversion of cloud droplets and accretion of cloud droplets by rain) with increased CDNC can be observed. This even leads to a complete shut-down of warm rain production in the Thompson scheme, which is also evident in the absence of rain hydrometeors in Fig. 4. As a result, evaporation in the lowest model levels decreases strongly for the high CDNC value in the simulations with the Thompson scheme. Both microphysics schemes show a significant decrease in the total amount of melting of frozen hydrometeors below the melting line at about 4 km height. The strong cooling due to evaporation and melting in the cleanest cases for the simulations with the Thompson scheme (Fig. 6 c) can explain the significantly shorter lifetime of the cell compared to the more polluted cases and the other bulk scheme. The dominant region of freezing processes is lifted from around 8 km height in the low CDNC case to around 10 km for the high CDNC case height in both schemes. While deposition on ice hydrometeors is a significant process for all values of CDNC for the Morrison scheme, it becomes more enhanced for the most polluted simulation using the Thompson scheme, related to the change in the dominant ice-phase hydrometeor class to snow (Fig. 5). Condensation onto cloud droplets is present in all simulations up to 10 km height in comparable amounts and dominates the latent heating due to the large energy transfer involved. Deposition processes onto ice hydrometeors are significant for both the cleanest and the most polluted simulation in the Morrison scheme, while the Thompson scheme shows much more deposition in the most polluted case, which can be related to the changes in the hydrometeor composition (Fig. 5). The decrease in the total amount of microphysical mass transfer in all simulations around 55 minutes into the simulations is caused by the splitting of the tracked cell into two individual cells. However, no significant change to the relative proportions of the different processes can be observed at this stage.

A more detailed analysis of the processes involved in the formation of rain over the lifetime of the cell in the different cases (Fig. 7) reveals that autoconversion of cloud droplets to rain for the highest CDNC values in both bulk schemes is almost negligible, with only very little autoconversion in the Morrison scheme, even for the smallest CDNC value. Accretion of cloud

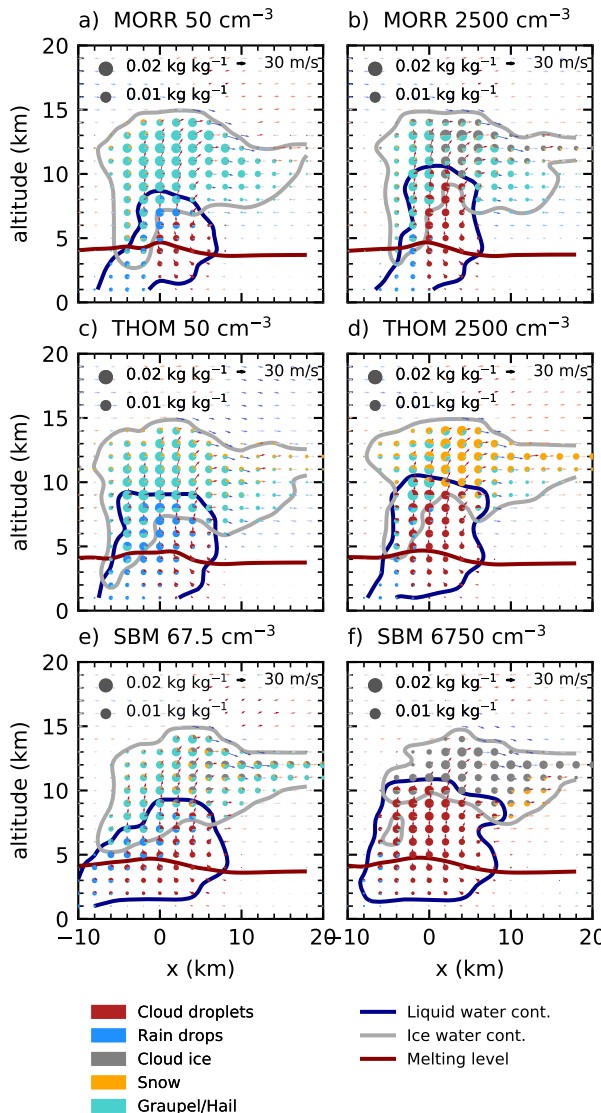

**Figure 5.** Hydrometeor mass mixing ratios in a slice along the line of travel of the cell for the cleanest (left) and most polluted (right) simulations after 60 minutes of simulation for the three microphysics schemes in CASE1.

droplets by rain is strongly depressed for high CDNC in both microphysics schemes. Melting of ice hydrometeors contributes significantly to the production of rain in both bulk schemes and is reduced for the high CDNC case, especially in the Thompson scheme.

The processes transforming liquid to frozen water can be further broken down into processes representing the freezing of indi-

5 vidual cloud droplets or raindrops and riming processes, in which existing ice-phase hydrometeors accrete liquid water (Fig. 8). For both bulk microphysics schemes, freezing of raindrops and cloud droplets occur in two separate layers, with freezing of

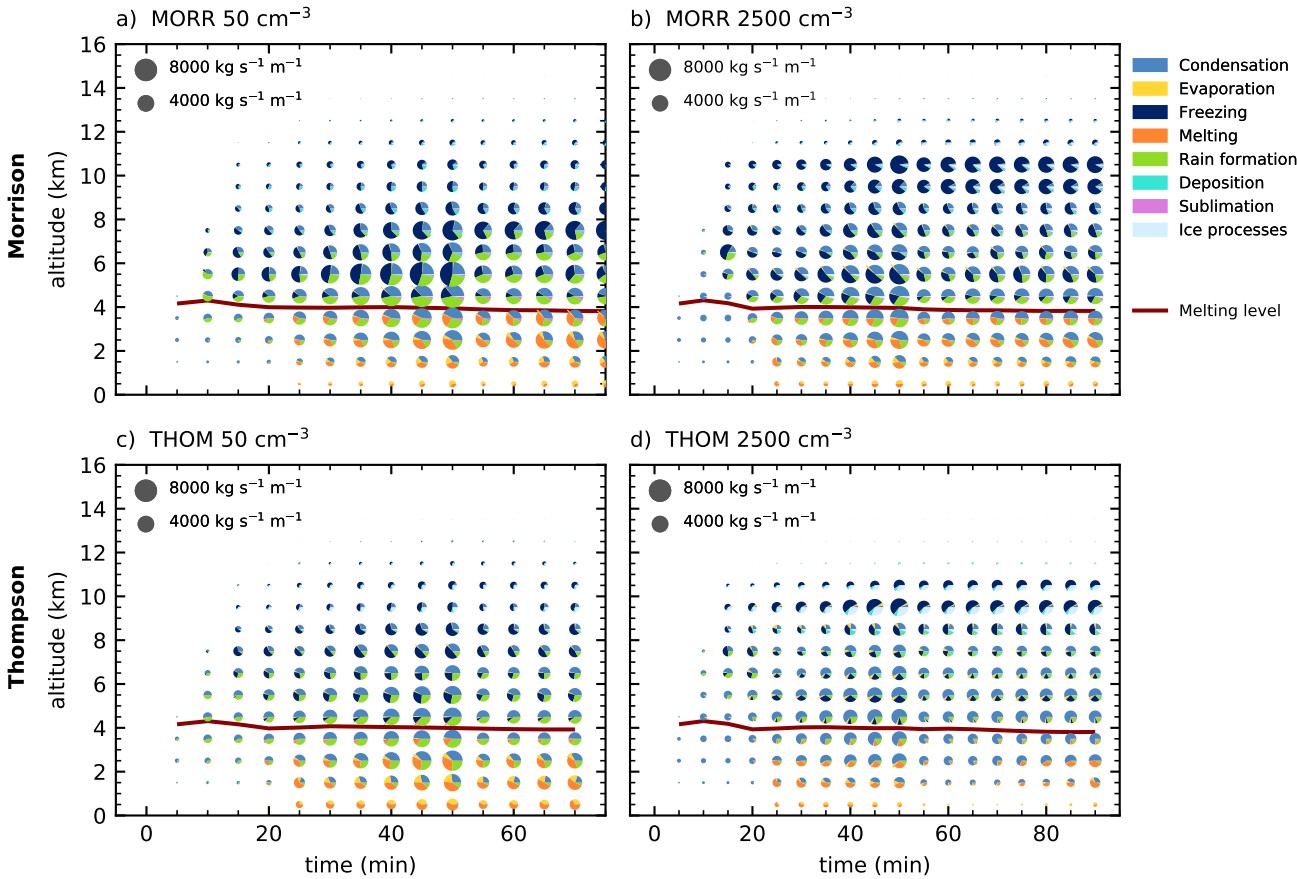

**Figure 6.** Time evolution of the microphysical process rates for the cleanest (left) and most polluted (right) simulations and the two bulk microphysics schemes (Morrison: top, Thompson: bottom) in CASE1. The pie charts denote mass transfer summed up over the volume of the cloud in each altitude interval for the different groups of microphysical process rates with the area of each colour proportional to the mass transfer. The red line shows the height of the $0°$C isotherm.

raindrops at around $8\,\mathrm{km}$ and freezing of cloud droplets above a height of $10\,\mathrm{km}$ up to $14\,\mathrm{km}$. In both microphysics schemes, freezing of raindrops is strongly decreased for increased CDNC (Fig. 8 b,d), while freezing of cloud droplets is increased by about a factor of three. This is not related to the parametrisation of the freezing processes (described in more detail in appendix A2), which does not include any information about cloud droplet effective radius and raindrop effective radius through the number concentrations. Instead, these changes are purely a result of the shift in the abundance of cloud droplets and raindrops (Fig. 5).

The riming processes are spread out over a much larger altitude range in the cloud, between the melting level at about $4\,\mathrm{km}$ and about $11\,\mathrm{km}$ height for riming of cloud droplets and below $9\,\mathrm{km}$ for the riming of raindrops. Riming is significantly stronger at all CDNC values in the simulations with the Morrison scheme (Fig. 8 a,b). In the Morrison scheme, riming of rain droplets

is strongly decreased for higher CDNC and mainly restricted to around 5 km height. In the Thompson microphysics scheme (Fig. 8 c,d), raindrop riming is also strongly decreased for high CDNC, but still occurs over the same height range as in the low CDNC case. Both microphysics schemes show a slight increase in droplet riming with higher CDNC over the entire altitude range. We can thus explain the shift in freezing and riming processes observed in Fig. 6 by a decreased riming of rain droplets

at lower altitudes and a shift from the freezing of raindrops to the freezing of cloud droplets occurring at higher altitudes.

The evolution of the deposition and sublimation processes (Fig. 9) shows substantial differences between the two bulk microphysics schemes and a strong response to a variation of CDNC. The calculation of deposition and sublimation in the microphysics scheme is explicitly parametrised for each hydrometeor class, taking into account detailed information on the size distribution of the hydrometeors (Thompson et al., 2004; Morrison et al., 2005). In the Morrison scheme (Fig. 9 a,b), the

increase in CDNC leads to a decrease of both deposition and sublimation over the entire height of the cloud. These processes dominantly occur on hail for the cleanest case and are more distributed over hail, snow and pristine ice in the polluted case, which agrees with the shifts in the hydrometeor mixing ratios (Fig. 5 a,b).

In the simulations with the Thompson microphysics scheme (Fig. 9 c,d), deposition and sublimation processes show very a different behaviour. The strong increase in snow in the cloud with increasing CDNC (Fig. 5 c,d) leads to a strong increase in

both deposition and sublimation on snow. Deposition on ice is on the same order of magnitude for the cleanest case, but not strongly affected by a change in CDNC. Sublimation of graupel only occurs around and below the melting layer and is significantly reduced by increasing CDNC. As deposition on graupel is prohibited in this microphysics scheme, there is no decrease in deposition on graupel associated with the changes in the hydrometeor ratio compensating the increase in deposition on snow. This leads to a strong increase in total deposition with increased CDNC as the main response in the Thompson scheme.

Latent heating constitutes a key feedback of the microphysics scheme onto the model dynamics along with changes to the buoyancy due to changes in condensate loading. The vertically resolved latent heating over the lifetime of the tracked cell in CASE1 is shown in Fig. 10 for all three microphysics schemes and split up into the individual phase changes for the two bulk microphysics schemes in Fig. 11.

Latent heat release from condensation is the dominant contribution to the latent heating and about a magnitude stronger than

the other contributions, thus determining the general shape of the latent heating profile (Fig. 10 and Fig. 11 a,g). The changes to condensation due to changes in CDNC in the two bulk microphysics schemes are comparatively small, which can be explained by the use of saturation adjustment in the calculation of the condensation, which does not include an effect of changes in droplet radius onto the condensation.

The same limitation applies to the evaporation of cloud droplets, which also cannot show any direct effect from changes in

CDNC due to the use of saturation adjustment. However, the evaporation shows much stronger differences between the two microphysics schemes and also a stronger effect of a variation in CDNC (Fig. 11 b,h). The strong changes in the evaporation at higher levels in the mixed-phase region of the cloud, especially for the Thompson scheme, can be explained with the changes in deposition on frozen hydrometeors (Fig. 11 e,k). The increased deposition with increasing CDNC through the changes to the frozen hydrometeors could lead to a further decrease of the saturation vapour pressure over water in the water-subsaturated

regions of the cloud and thus additional evaporation. There is also a noticeable decrease in condensation in the higher layers of

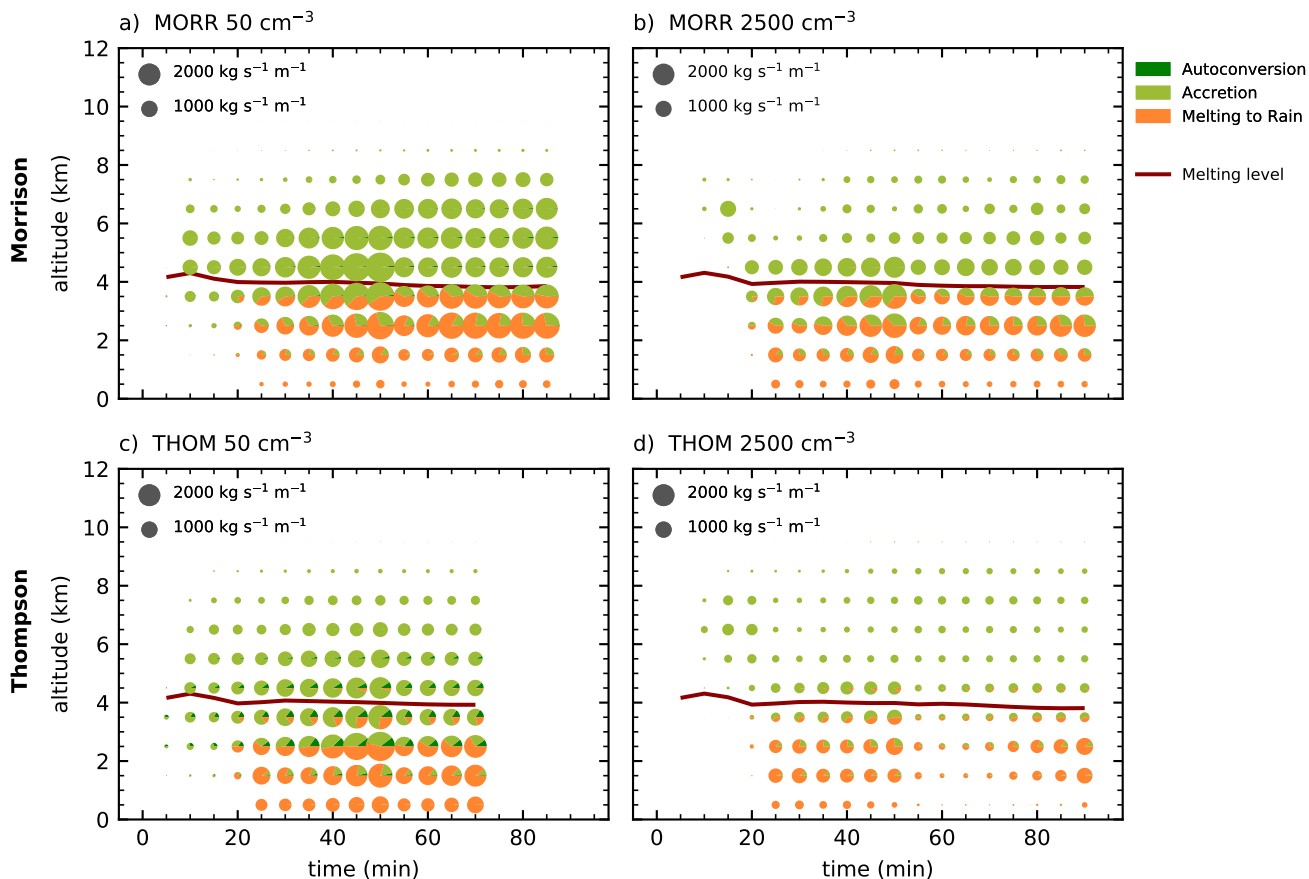

**Figure 7.** Time evolution of the microphysical process rates relevant for rain formation processes (autoconversion, accretion of cloud droplets by rain and melting of ice hydrometeors) as in Fig. 6.

the mixed-phase region of the cloud at around 10 km for the Thompson scheme (Fig. 11 g), which could be similarly related to the increase in deposition. The evaporation in the lower layers is associated with the evaporation of raindrops. The differences between the two schemes and the variation with changes in CDNC can be directly related to the differences in the amount of rain, which is both higher and more strongly decreasing with increasing CDNC in the Thompson scheme than in the Morrison

5     scheme.

All three microphysics schemes show a small shift of latent heating to higher altitudes superimposed on that in the range between $7\,\mathrm{km}$ and about $10\,\mathrm{km}$ for increasing CDNC (Fig. 10), which can be associated with the shifts in freezing and riming (Fig. 11 d,i), described in more detail in Fig. 8. The decrease in latent cooling from melting processes in the lowest layers is stronger in the Thompson scheme than in the Morrison scheme (Fig. 11 b,h).

10    There are large differences between the microphysics schemes in the latent heating and cooling from sublimation and deposition and its response to changes in CDNC. The Morrison scheme shows a significant decrease of both sublimation and

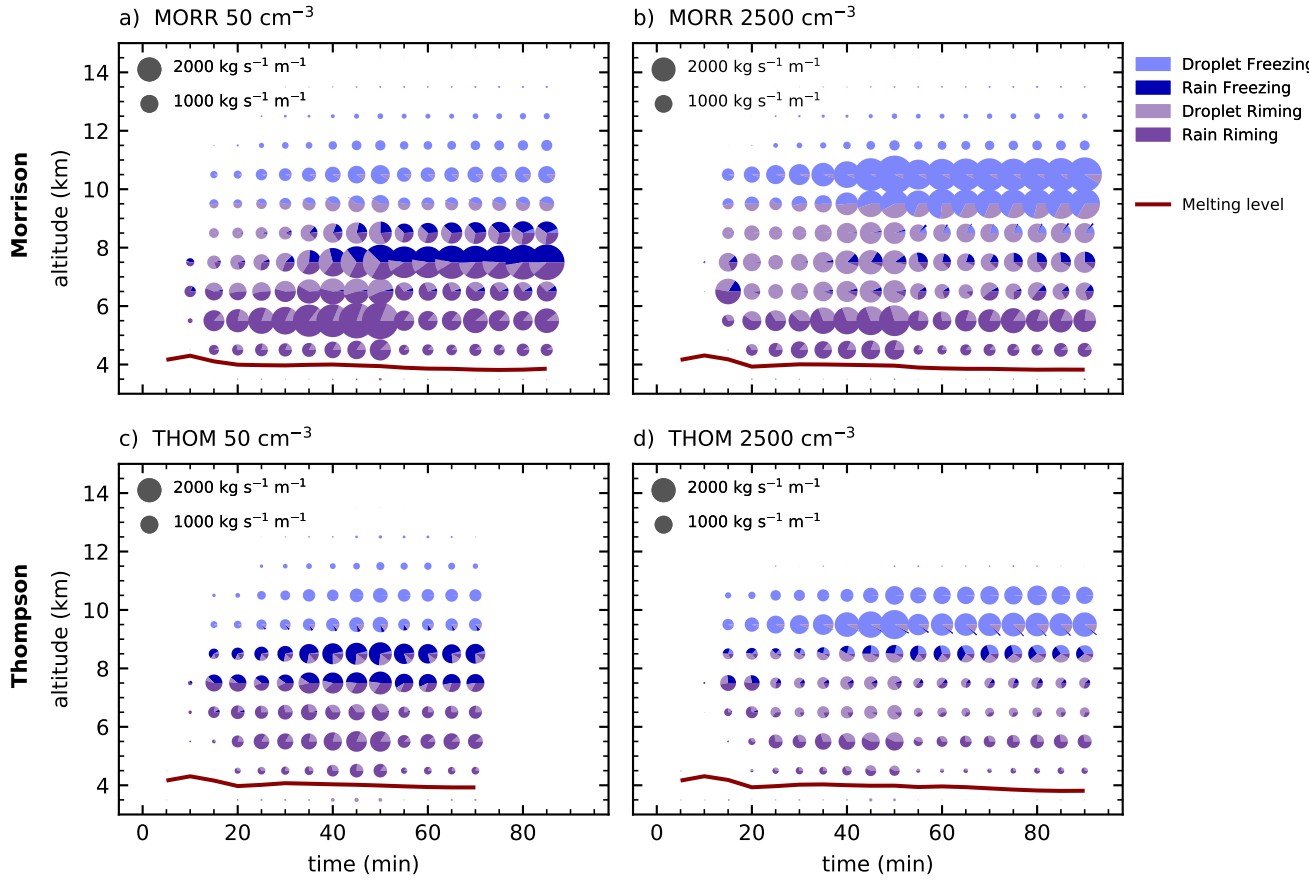

**Figure 8.** Time evolution of the the microphysical process rates of freezing and riming processes as in Fig. 6.

deposition with increased CDNC (Fig. 11 e,f). Apart from changes due to the shift in hydrometeors from hail to snow and cloud ice (Fig. 5 and Fig.9), these decreases can be related to the lower amount of ice hydrometeors in the mixed phase region of the cloud. Although these two changes cancel each other to a large extent in the integrated latent heating, the two processes occur at different heights, which results in a shift of latent heating to lower levels, opposing the changes to the freezing and

5  riming processes (Fig. 11 c). Furthermore, this strong decrease in sublimation leads to a decrease in water vapour near the cloud base, which could cause the consistent decrease in condensation at around 5 km altitude in the Morrison scheme (Fig. 11 a).

In the Thompson scheme, sublimation of ice hydrometeors is weak and barely affected by changes in CDNC (Fig. 11 l). However, increases in CDNC lead to an increase in deposition in the higher parts of the cloud (Fig. 11 k). This effect can be explained by the observed shift in hydrometeors from graupel to cloud ice and snow since deposition on graupel is turned

10  off in the Thompson microphysics scheme, while it occurs on both snow and cloud ice. This increase in deposition could be the main reason for the changes observed in evaporation of cloud droplets as it significantly increases the sub-saturation over water in the mixed phase in regions that are supersaturated with respect to ice. This can be interpreted as a manifestation of

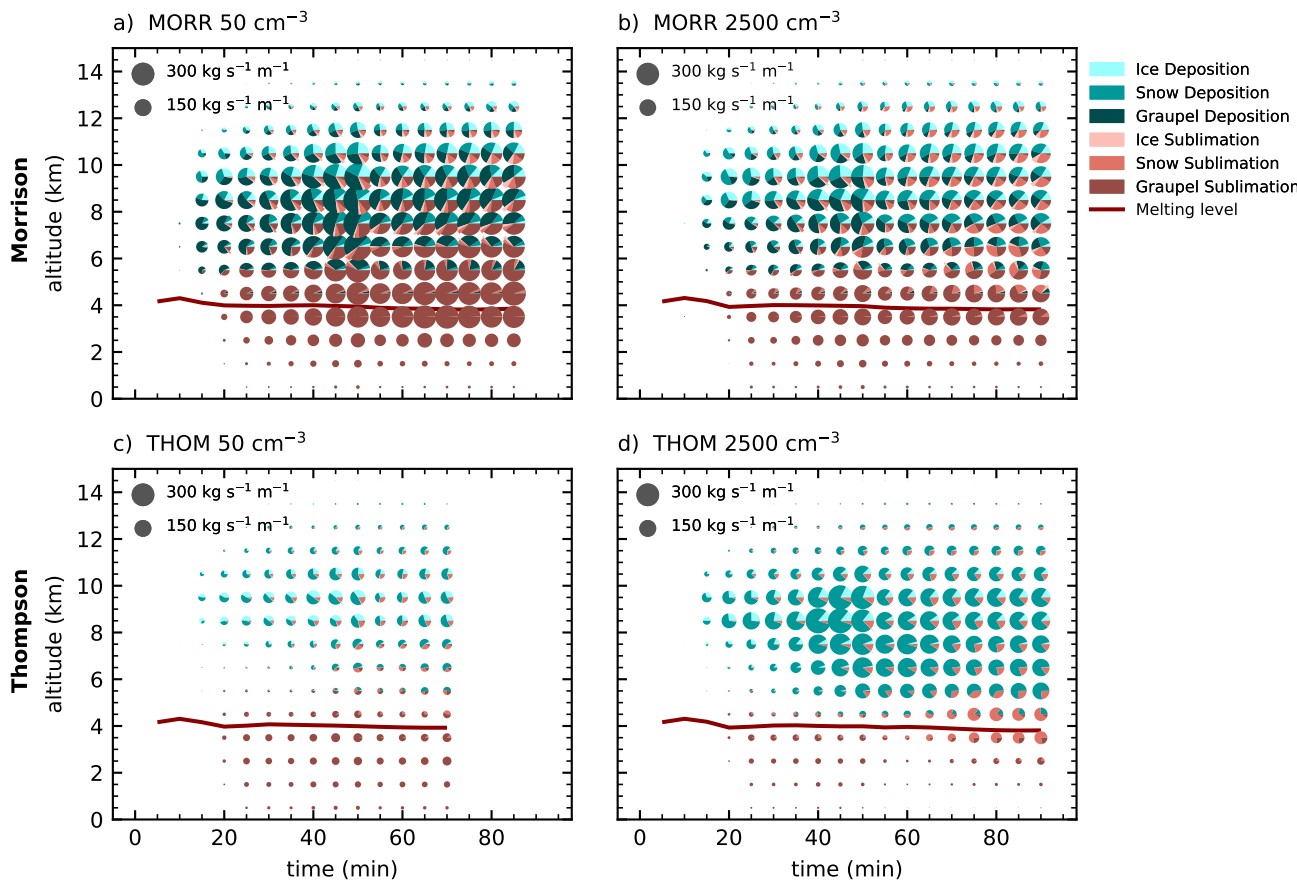

**Figure 9.** Time evolution of the microphysical process rates of deposition and sublimation as in Fig. 6.

the Wegener-Bergeron-Findeisen process (Wegener, 1911; Findeisen, 1938; Findeisen et al., 2015; Storelvmo and Tan, 2015), transferring water mass from liquid hydrometeors to the frozen hydrometeors. This constitutes an additional feedback from the changes in the ice phase back onto the liquid phase hydrometeors.

In contrast to the increased latent heating from freezing or melting, changes in condensation and evaporation, as well as in sublimation and deposition, are linked to a change in condensate loading, which affects the buoyancy of the cloud and thus at least partially buffers the impact of latent heating and cooling on the dynamics of the clouds.

The changes to the vertically integrated latent heating in the cloud for all three microphysics schemes do not show a significant trend with increasing CDNC (Fig. 10 d,e,f). The Thompson scheme shows lightly higher integrated latent heating for the two simulations with the highest CDNC content, but no consistent trend over the rest of the simulations (Fig. 10 e). The SBM simulations show a slightly decreasing trend of integrated latent heating for the highest CDNC values above $1000\,\mathrm{cm}^{-3}$ but no consistent trend over the entire range of values (Fig. 10 f) . Despite the significant change to the altitude of freezing there is no systematic change in the integrated latent heat release from freezing for both bulk microphysics schemes that would

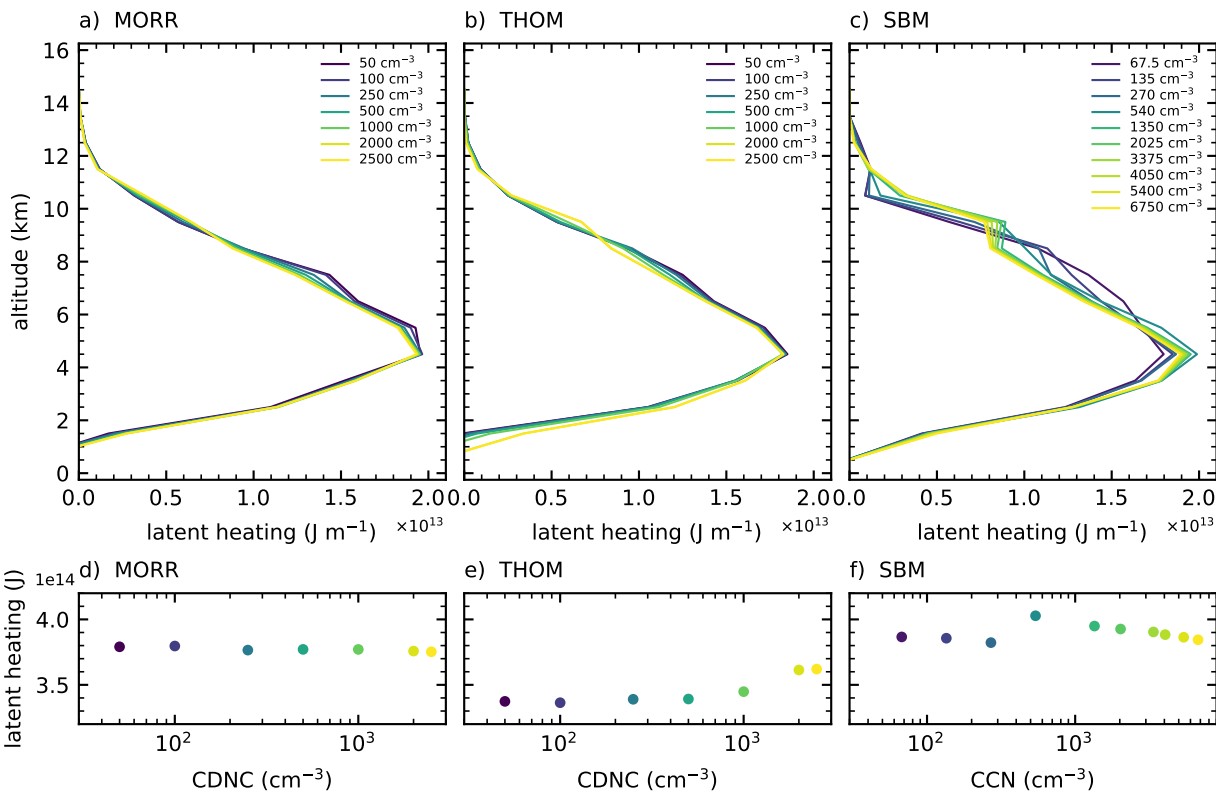

**Figure 10.** Profiles of the sum of latent heating over the lifetime of the dominant tracked cell for the three microphysics schemes in CASE1.

contribute to an invigoration of the cloud. In the Morrison scheme, the strong changes in deposition and sublimation almost entirely cancel out when integrated vertically. In the Thompson microphysics scheme, the increase in the integrated latent heat release from deposition cancels out the significant decrease in the integrated evaporation of cloud droplets and rain.

### 3.3 Effects on cloud mass and centre of gravity

The tracking and watershedding allow for a determination of the cloud mass inside the identified cloud volumes and the centre of gravity of the hydrometeors in the cloud. These analyses are also performed separately for the liquid-phase and ice-phase hydrometeors in the cloud, which allows us to relate the changes in the properties for the entire cloud to changes in the individual phases.

The evolution of the cloud mass and the mass of the two water phases in the cloud (Fig. 12) in the three microphysics schemes is similar, with a maximum cloud mass of about $2 \cdot 10^{10}$ kg for all microphysics schemes before the splitting of the cell and then

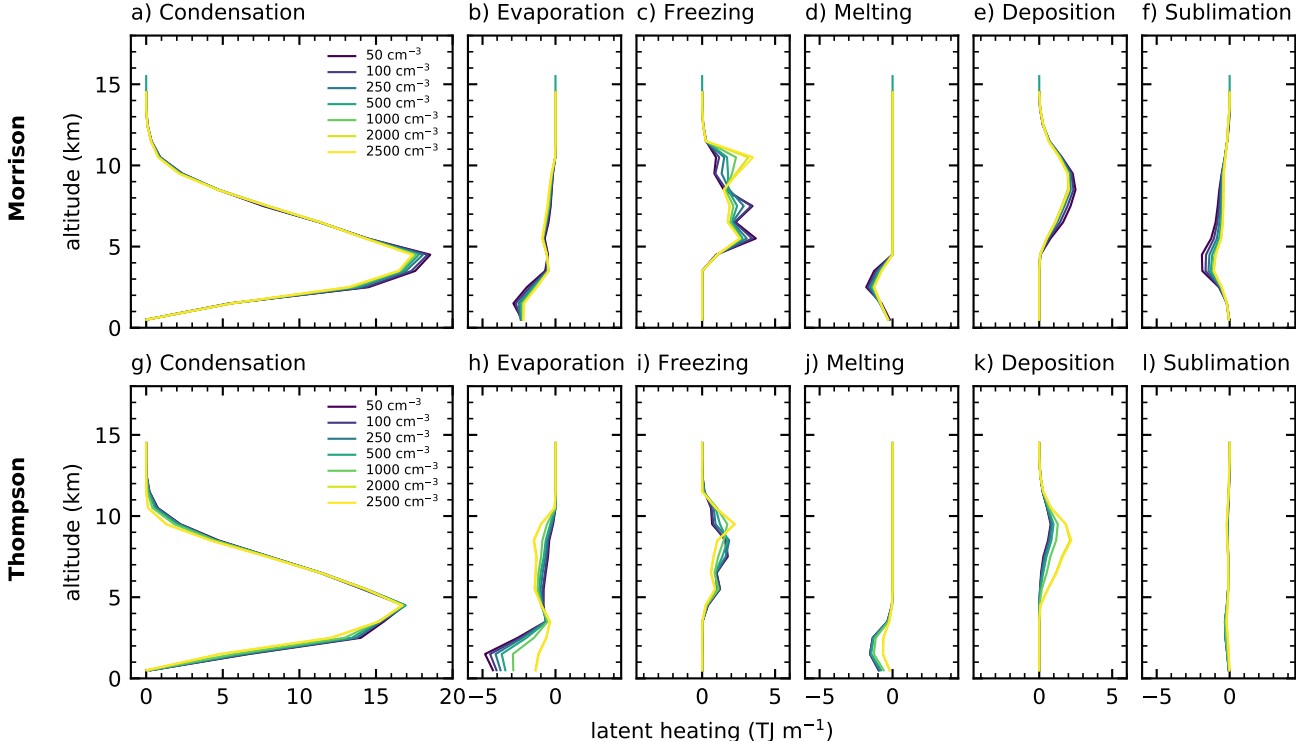

**Figure 11.** Profiles of the components of the latent heating and cooling over the lifetime of the tracked cell for the two bulk microphysics schemes in CASE1.

about $1.5 \cdot 10^{10}$ kg for the two bulk microphysics schemes (Fig. 12 a,b) and slightly higher cloud masses of up to $1.8 \cdot 10^{10}$ kg in the spectral bin microphysics scheme (Fig. 12 c). The cloud mass and also the difference between the bulk schemes and the bin scheme are dominated by the ice-phase hydrometeors, while the liquid-phase mass is very similar in all three different microphysics schemes, making up about 20-25% of the total cloud mass.

5    There are, however, marked differences in the response to changes of the aerosol proxy between the microphysics schemes. The Morrison scheme shows a decrease of total cloud mass and ice-phase mass by about 10-15% over the range in which we increase the CDNC and no significant changes in the liquid phase. This decrease in ice-phase mass can be directly linked to the changes in the microphysical process rates analysed in Sec. 3.2. The shift of freezing to higher altitudes leads to a reduction in frozen hydrometeors in the mixed phase of the cloud and thus significantly less growth of the ice phase through vapour

10   deposition. In the Thompson scheme, however, increased CDNC leads to an increase in ice-phase and total mass and a small increase in cloud liquid mass. This increase agrees well with the increased deposition due to the changes in the ice hydrometeor partition in the cloud discussed in Sec. 3.2. In the simulations using the SBM scheme, the two phases show a differing response to the aerosol proxy with increased liquid hydrometeor mass and a decrease in ice-phase mass for increasing CCN.

The altitude of the centre of gravity is affected by the choice of microphysics scheme, with an overall higher centre of gravity

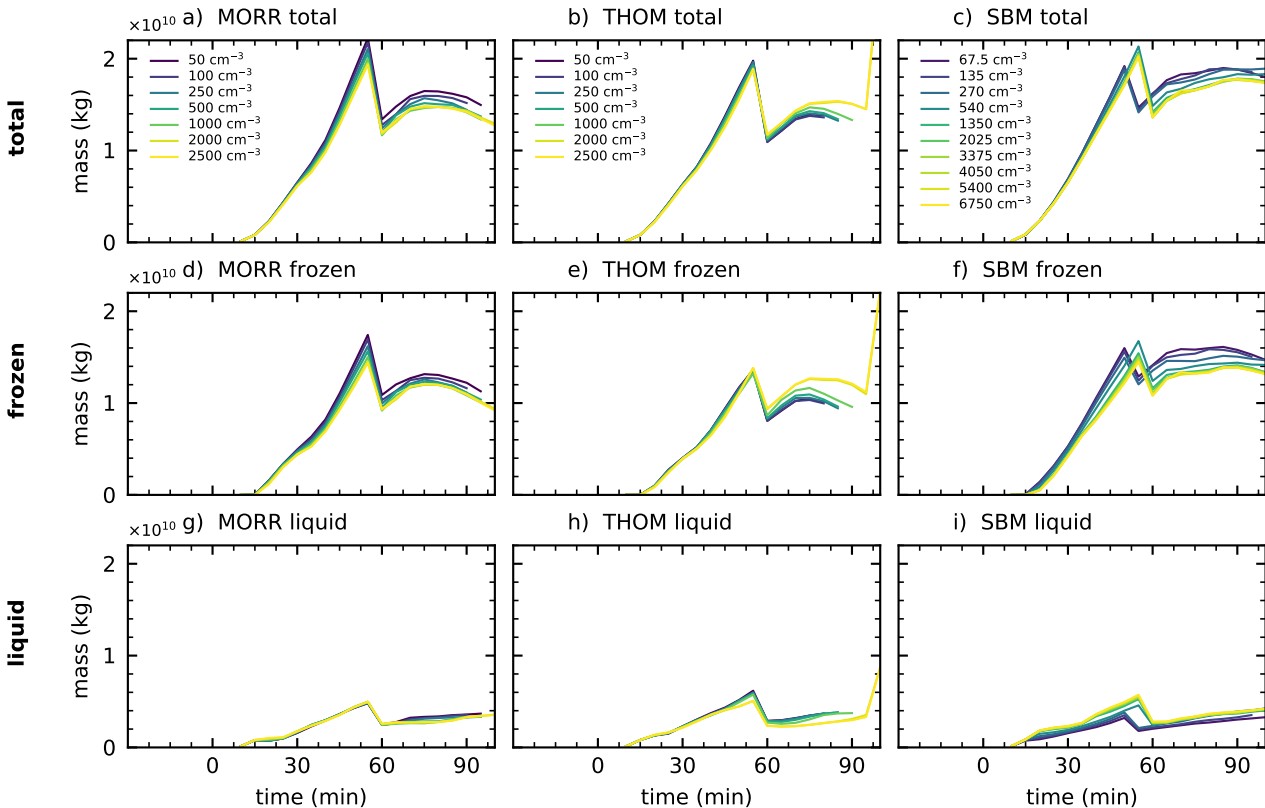

**Figure 12.** Total water mass, liquid water mass and frozen water mass in the analysed right-moving cell for the three different microphysics schemes (Morrison: left, Thompson: middle, SBM: right) in CASE1. The jump in the curves occurs at the point where the cell splits into two individual cells.

for the SBM scheme (Fig. 13 c) compared to the two bulk microphysics schemes (Fig. 13 a,b).

There is a consistent response in the cloud heights for all three microphysics schemes. The microphysics schemes show an increase in the height of the centre of gravity of the entire cloud, which is more pronounced using the Thompson scheme (about 1.5 km) than in the Morrison scheme(about 0.5-1 km). This includes an upward shift in both the liquid and frozen water
5    in the cloud. The increased height of the liquid phase can be directly related to the decrease in the formation of warm rain (Fig. 6) and the more numerous cloud droplets reaching higher up in the cloud in the polluted case compared to the dominating raindrops in the cleanest case (Fig.5). The increase in the altitude of the ice phase in the cloud with increased CDNC can be related to the changes in the altitude of the freezing processes. However, it can also be a result of the lower fall speeds of the ice and snow hydrometeors dominating in the polluted case instead of graupel and hail in the cleanest cases. As for the bulk
10    microphysics schemes, there is an increase in the height for both phases in the simulations using the SBM scheme, which is significantly more pronounced in the liquid phase of the cloud.

All three microphysics schemes show a clear saturation in the effect of changes in the CDNC/CCN concentration. Variations

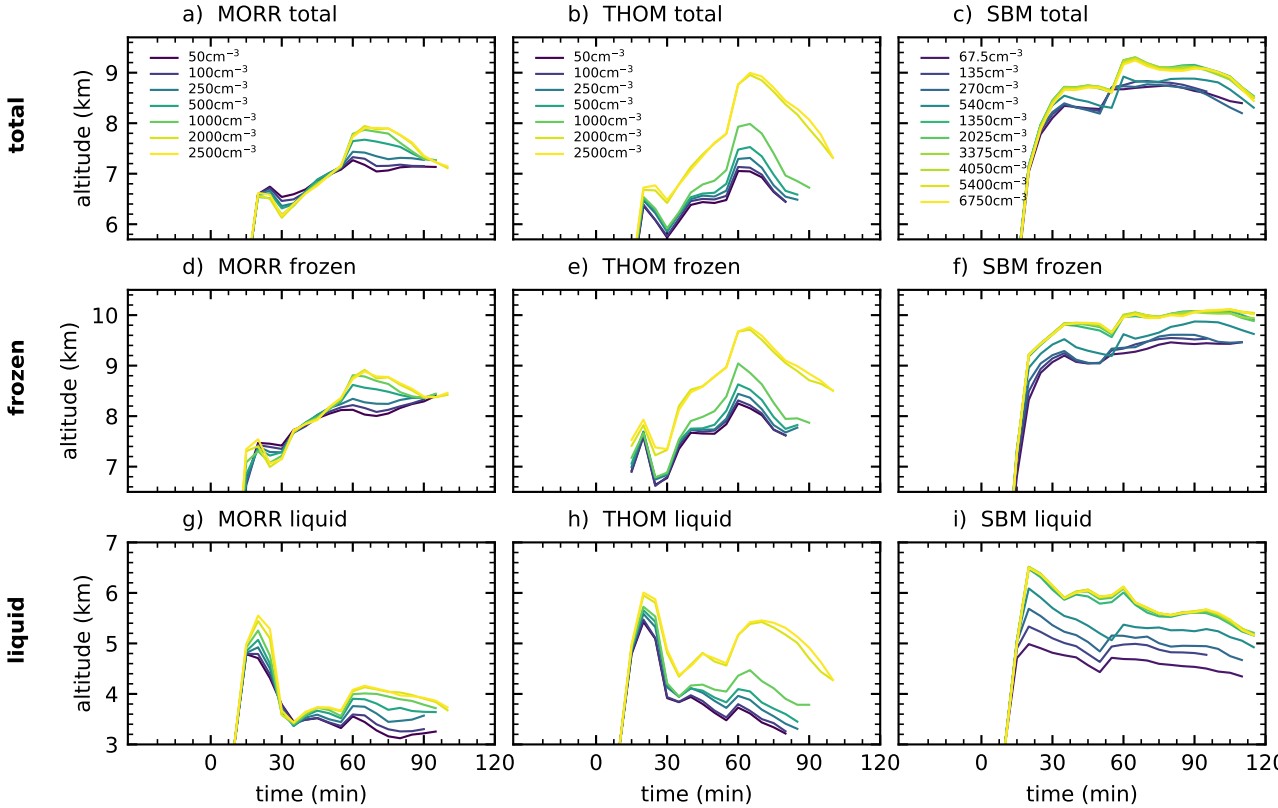

**Figure 13.** Altitude of the centre of gravity of the cloud and the individual phases in the analysed right-moving cell for the three different microphysics schemes (Morrison: left, Thompson: middle, SBM: right) in CASE1.

above $2000\,\mathrm{cm^{-3}}$ in the bulk schemes and above $1350\,\mathrm{cm^{-3}}$ in the SBM simulations only lead to insignificant effects on both the cloud mass and the altitude of the centre of gravity of the different phases.

### 3.4 Sensitivity test: a second idealised supercell case (CASE2)

To investigate the representativeness of the results and the response of the deep convective clouds to the variation of aerosol

5 proxies CDNC and CCN, the same set of simulations and analyses have been performed for a second idealised supercell case (CASE2) with different forcing and initial conditions (Section 2.1).

The time evolution of the cloud averaged process rates for the two bulk microphysics schemes (Fig. 14) shows that the total microphysical water transfer is much weaker in CASE2 than in CASE1, with process rates about a factor of three smaller. This case shows much stronger differences between the two bulk microphysics schemes in the general evolution of convection. For

10 the Morrison microphysics scheme, a development of the convective cloud in two stages occurs. After an initial maximum in the microphysical processes after around 30 minutes of simulation time, the convective activity becomes weaker before picking

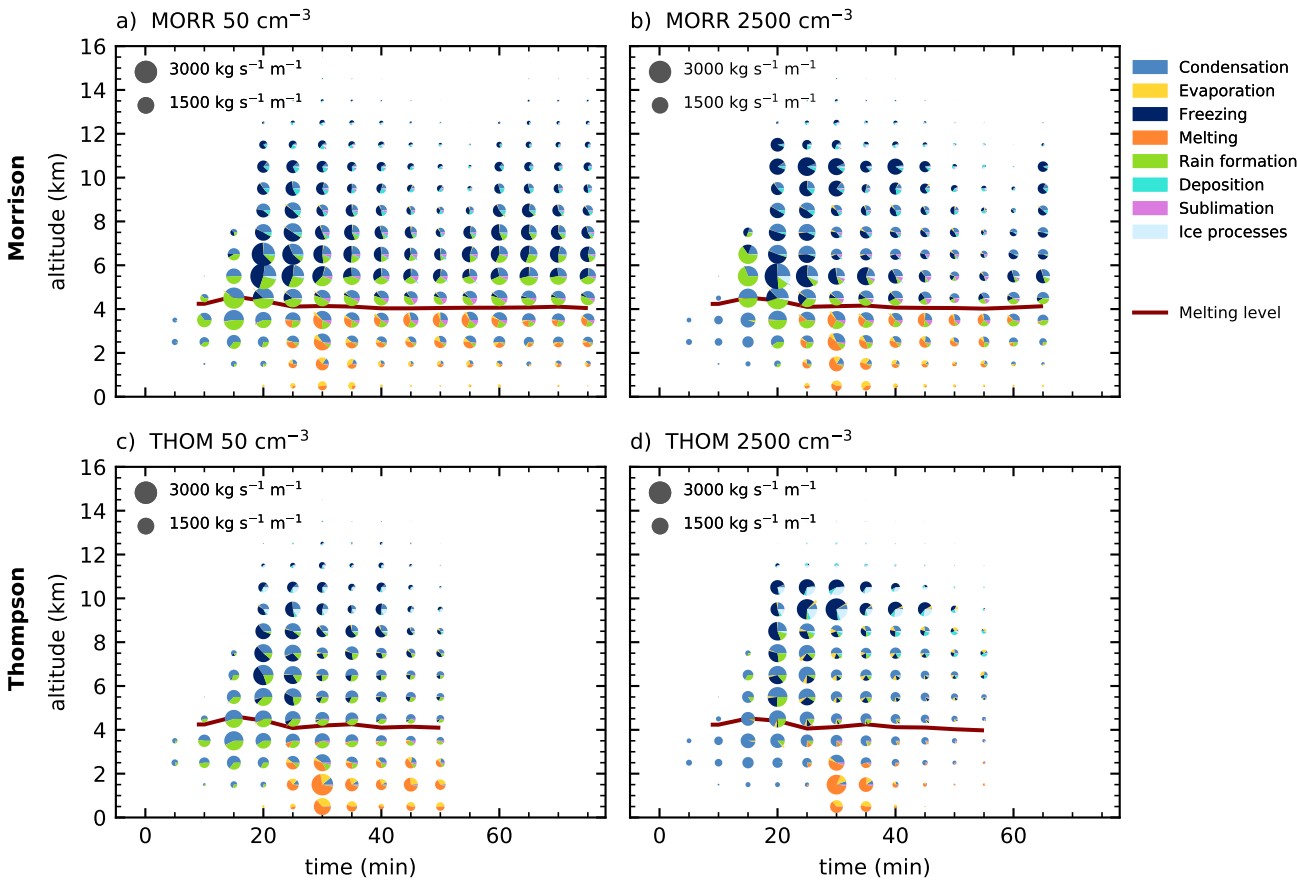

**Figure 14.** Temporal evolution of the microphysical process rates in CASE2 for the cleanest (left) and most polluted (right) simulations and the two bulk microphysics schemes (Morrison: top, Thompson: bottom). The pie charts denote the different groups of microphysical process rates with the area proportional to the sum of the microphysical process rates in the specific altitude interval inside the cloud volume.

up again after about an hour of simulation time. For the Thompson microphysics scheme, this second episode of development in the tracked cell is completely absent for all simulations, with the cloud dissipating after about 60 minutes of simulation time. This is potentially related to the substantially higher cooling at and below cloud base due to the evaporation of rain and the melting of frozen hydrometeors. The cooling can substantially weaken the convective updraft and thus prevent the further

5   development of the cell that takes place in the simulations using the two other microphysics schemes. This finding agrees with a substantially shorter lifetime of the cleanest case for the simulations with the Thompson scheme in CASE1 (Fig. 6).

Despite these differences in the evolution, CASE2 shows very similar changes in the microphysical processes due to a variation of CDNC to CASE1 for both microphysics schemes. The formation of rain due to autoconversion of cloud droplets and accretion by rain is smaller and restricted to lower heights in the polluted case using the Morrison microphysics scheme. For

10   the Thompson microphysics scheme, the formation of rain is decreased and shifted to higher levels in the model under polluted

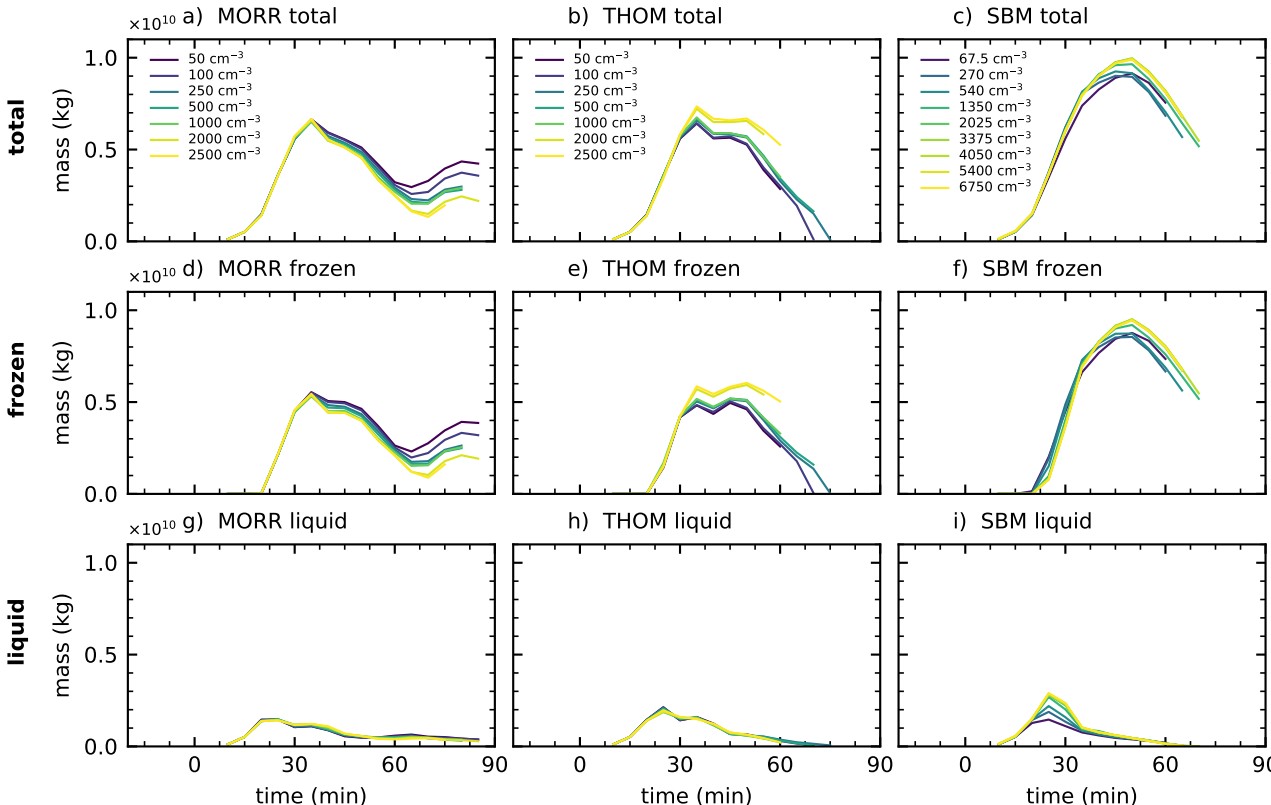

**Figure 15.** Total water mass, liquid water mass and frozen water mass in the analysed left-moving cell for the three different microphysics schemes (Morrison: left, Thompson: middle, SBM: right) in CASE2.

conditions. Furthermore, the freezing and riming processes predominantly occur at higher altitudes than in the clean case for both bulk microphysics schemes.

In line with these changes to the microphysical process rates, the evolution of the cloud mass in CASE2 (Fig. 15) is smaller than in CASE1 for the two bulk microphysics schemes, with about half as much hydrometeor mass in the cloud up to about

5    $5 \cdot 10^9$ kg. The ice phase is more dominant, with the liquid phase of the cloud only accounting for less than a quarter of the total cloud mass. The simulation with the spectral bin microphysics scheme shows a larger cloud mass than the two bulk schemes for this case, only about 30% smaller than in CASE1 (Fig. 15 a,b,c), which includes much more frozen hydrometeor mass than the two bulk microphysics schemes (Fig. 15 d,e,f), while liquid-phase mass is similar between the three microphysics schemes (Fig. 15 g,h,i).

10    The effects of a variation of CDNC are quite similar to the ones seen in CASE1 for the two bulk microphysics schemes (Fig. 15 a,b). The simulations with the Morrison scheme show a relatively small decrease in cloud mass, while cloud mass increased by about 15% for the Thompson microphysics scheme. These changes are almost entirely due to changes in the

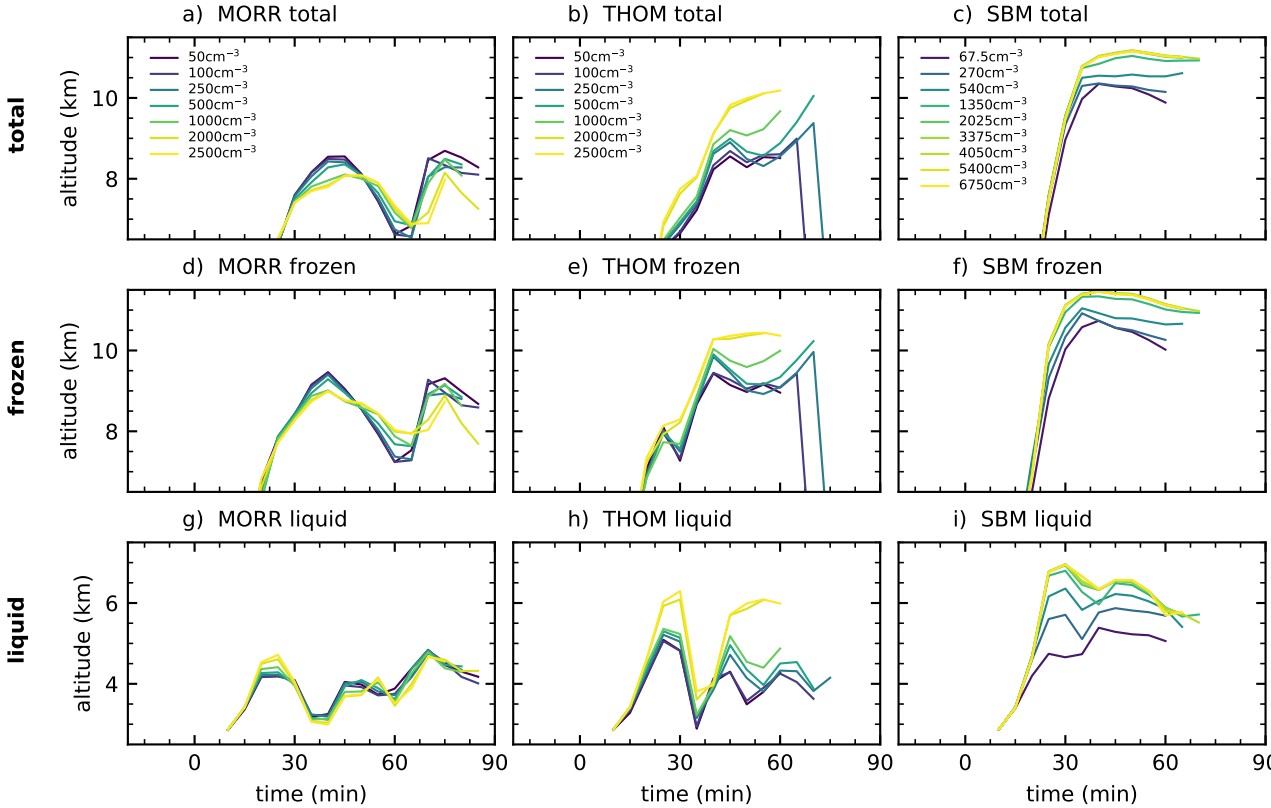

**Figure 16.** Altitude of the centre of gravity of the cloud and the individual phases in the analysed left-moving cell for the three different microphysics schemes (Morrison: left, Thompson: middle, SBM: right)in CASE2.

ice phase of the clouds with insignificant effects of a variation in the liquid phase (Fig. 15 g,h) for both bulk schemes. The simulations with the spectral bin microphysics scheme, however, show an opposite response compared to CASE1 with an increase of cloud mass of a similar magnitude as the variation in the two bulk microphysics schemes (Fig. 15 c), which is dominated by changes in the ice phase (Fig. 15 f). There is a significant increase of almost 50% in cloud liquid mass in the
5   earlier stages of the cloud evolution (Fig. 15 i) at around 25 minutes of simulation time between the cleanest and the most polluted simulation with the SBM scheme. This coincides with a delayed evolution of the ice phase during that period of the developing cloud.

The changes in the altitude of the centre of gravity show less clear relationships to changes in the aerosol proxies CDNC/CCN in this case for the two bulk microphysics scheme. The Morrison scheme (Fig. 16 a,d,g) has the strongest variation in the time
10   evolution of the altitude of the centre of gravity but generally shows a decrease of the altitude for both the liquid and the ice phase in the cloud. In the Thompson scheme (Fig. 16 b,e,h) increased CDNC leads to an increase in the height of the centre of gravity of the entire cloud and of both phases of water in the cloud. Similarly, increasing CCN in the spectral-bin microphysics

scheme (Fig. 16 c,f,i) leads to a strong increase in the altitude of the cloud mass and the individual phases, with the COG of total mass about 1.5 km higher in the most polluted case (6750 cm$^{-3}$) compared to the clean case (67.5 cm$^{-3}$) and even stronger shift of up to 2 km in the liquid phase. All the SBM simulations with a higher CCN value than about 1500 cm$^{-3}$ lead to relatively similar results, which means that the aerosol effects saturate at this value.

## 4   Conclusions

We investigated the effects of changes in cloud droplet number concentration (CDNC) and cloud condensation nuclei (CCN) concentrations on the development of idealised simulations of deep convection to test proposed aerosol effects. This includes different mechanisms of convective invigoration (Rosenfeld et al., 2008; Lebo and Seinfeld, 2011; Fan et al., 2013; Grabowski and Morrison, 2016). A combination of cell tracking and detailed process rate diagnostics were used to investigate the evolution
and structure of the microphysical processes in individual deep convective cells. We used three different cloud microphysics schemes (two bulk schemes and one bin scheme) to investigate how the choice of microphysics scheme affects these results. By covering a wide range of values of CDNC/CCN representative of conditions from very clean to very polluted, we were able to look for consistent responses of the clouds to changes in these aerosol proxies and thus go beyond a simple comparison of just clean and polluted conditions.

An increase in cloud droplet number concentration from values representing clean conditions (CDNC=50 cm$^{-3}$) to strongly polluted conditions (CDNC=2500 cm$^{-3}$) leads to a shift of freezing processes to higher levels in both bulk microphysics schemes. Detailed analyses of the individual process rates confirmed that this is indeed related to a shift from freezing of rain to freezing of cloud droplets and a decrease in riming of raindrops due to larger amounts of liquid water in the form of cloud droplets instead of rain. This, in turn, can be related to the changes in autoconversion and accretion in the warm-phase region
of the cloud. This is in line with the first step of the mechanisms proposed for convective invigoration of deep convection due to an increase in aerosols acting as CCN (e.g. Rosenfeld et al., 2008; Lebo and Seinfeld, 2011; Fan et al., 2013; Altaratz et al., 2014). These changes are concurrent and linked to changes in the prevailing hydrometeors in the different parts of the clouds. Both bulk microphysics schemes showed a strong increase in cloud droplet mass mixing ratio at the expense of raindrops for increased CDNC. This shift leads to a significant increase of the height of freezing and riming processes, which shifts the latent
heat release from freezing upwards by about two kilometres. This response is consistent between the different microphysics schemes and confirms earlier studies that stated the importance of changes in the partition between rain and cloud droplets in determining the evolution of freezing and riming (Seifert and Beheng, 2006; Kalina et al., 2014). The simulations with the SBM scheme show an upward shift in latent heating that is very similar to the one observed for the two bulk schemes and associated with the lifting of the freezing and riming processes. This confirms that the effect is not just an artefact of the separate
treatment of raindrops and cloud droplets in the bulk microphysics schemes or the application of saturation adjustment. In the ice phase of the clouds, there is a clear shift from mainly graupel or hail in the low-CDNC simulations to larger fractions of snow and ice crystals in the high-CDNC simulation.

A more detailed analysis of the different components of the latent heating for the two bulk microphysics schemes shows a

complex superposition of changes to the different phase changes in the tracked cells. This confirms results from previous studies on the effects of aerosols on supercells (Khain et al., 2008; Morrison, 2012; Kalina et al., 2014) and other deep convective clouds (Ekman et al., 2011) that pointed out a range of compensating processes limiting convective invigoration and a strong dependency on the environmental conditions in which the cloud develops. Condensation and evaporation are the largest contributions to latent heating and cooling in the cloud. The relative changes in these two processes due to changes in the aerosol proxies are comparatively small, except for the changes in the evaporation of rain due to the strong decrease in the formation of rain. This is to be expected, as condensation and evaporation of cloud droplets in the two bulk microphysics schemes are represented using saturation adjustment, which does not include the effect of changes in cloud drop radius on the condensation and evaporation processes. Saturation adjustment has the potential to mask the effects of aerosols in highly supersaturated strong convective updrafts as described, e.g. in Lebo et al. (2012) and Fan et al. (2018). Lebo et al. (2012) argue that saturation adjustment, as used in both bulk microphysics schemes in this study, leads to an artificial increase in condensation in the lower levels of the clouds, which would limit the effects of aerosol concentrations on buoyancy in mid and high levels.

There are significant differences between the two bulk schemes in the profiles of sublimation and deposition as well as in the response of these processes to changes in CDNC. This can be attributed to different parameter choices in the schemes. The strongest differences result from the fact that deposition onto graupel hydrometeors is not allowed to occur in the Thompson microphysics scheme, which leads to a strong increase in deposition due to the replacement of graupel by the other ice-phase hydrometeors on which deposition occurs. This strong increase in deposition additionally drives changes in condensation and evaporation in the mixed-phase region of the cloud via the Wegener-Bergeron-Findeisen process. By effectively removing water vapour, this leads to a noticeable feedback on the evaporation and condensation on cloud droplets that are intrinsically not affected by changes in CDNC because of the use of saturation adjustment. It was also shown that the melting of frozen hydrometeors contributes significantly to the formation of raindrops, especially under high CDNC conditions, which forms an additional important feedback of changes in the ice-phase onto the warm-phase processes.

The changes to the individual components of integrated latent heating in the cloud due to a variation of CDNC compensate each other in the two bulk microphysics schemes. Hence, there is no significant change in the total integrated latent heating in the cloud with changes in CDNC/CCN and no thermodynamic invigoration from changes in the microphysics due to the change in the aerosol proxies. This result is confirmed in the SBM simulations, that also do not show any significant change in vertically integrated latent heating for a variation of CCN. Therefore, the absence of convective invigoration in the bulk microphysics schemes cannot be solely attributed to the application of saturation adjustment.

The analysis of the clouds with respect to the total cloud mass and the altitude of the centre of gravity showed some contrasting results between the different microphysics schemes. There is a clear signal of a lifting of all parts of the clouds to higher altitude under polluted conditions, probably associated with the changes in the ice-phase hydrometeor partition. This agrees with findings from, e.g. Fan et al. (2013), reporting substantial changes to cloud height and even in the absence of convective invigoration in the form of increased total latent heating in the cloud. However, the analysis of cloud mass revealed opposing trends in the response between the three microphysics schemes. There is no clear pattern in the different responses to CDNC/CCN with regard to these bulk cloud properties, with variations between the two bulk microphysics schemes often as

large as between the bulk schemes and the spectral bin microphysics scheme, which confirms the strong differences between microphysics schemes found in previous studies (Lebo et al., 2012; Khain et al., 2015; White et al., 2017).

The results for the first case (CASE1), based on Weisman and Klemp (1982), are supported by the analysis of a second idealised supercell case (CASE2), based on Kumjian et al. (2010); Dawson et al. (2013). The microphysical process rate diagnostics re-
vealed similar changes in rain formation and the altitude of freezing and riming processes for the two bulk microphysics schemes in this second case. All three microphysics schemes showed that the effects of a variation of CDNC or CCN saturate above a threshold value in both simulated cases. Variations above a CDNC of around $2000\,\mathrm{cm}^{-3}$ in the bulk schemes and above a CCN concentration of $1500\,\mathrm{cm}^{-3}$ in the bin microphysics scheme do not lead to any further changes in the convective clouds with regard to cloud condensate mass or altitude. This confirms results from previous studies such as Kalina et al. (2014) that
reported a saturation of aerosol effects at similar values.

The pathway analysis developed for this study also includes the process rates for the number concentrations of the different hydrometeors. This includes processes like ice multiplication that could play an important role to better understand some of the possible pathways of aerosol effects on convective clouds (Fan et al., 2013, 2016).

This work focused on the analysis of microphysical pathways of aerosol effects on deep convective clouds in an idealised
framework. To test the robustness of the results under realistic scenarios, including potential buffering mechanisms, we are currently applying our analysis framework to large case study simulations of isolated convection over the area around Houston, Texas as part of the ACPC initiative (Aerosol, Cloud, Precipitation, and Climate Working Group, http://www.acpcinitiative.org) . We apply the cell tracking algorithm and the analysis of the detailed process-rate output developed in this study for a range of different cloud resolving models and contrasting aerosol conditions. In these simulations, the individual deep convective clouds
in the cloud field evolve and interact freely, which allows for a thorough analysis of important aspects such as the impact of aerosol conditions on the cell lifetimes or the statistics of the cloud size spectrum. The introduction of parameters describing the entire convective cell such as cloud mass and the position of the centre of gravity can contribute to a meaningful analysis cloud field simulations with a large number of individual clouds.

The understanding of the detailed structure of microphysical processes in individually tracked deep convective clouds and the
analysis of the pathways through which aerosol perturbations affect the deep convective clouds advances our understanding of aerosol-cloud interactions. This can be used to inform the parametrisation of microphysical processes and aerosol-convection interactions in global climate models. Recent developments in the use of global cloud resolving models in climate research (e.g. Ban et al., 2014; Seiki et al., 2014; Sato et al., 2018) further motivate a detailed understanding of the pathways of aerosol effects on convective clouds and the uncertainties in their representation in numerical models.

## A1   Convective cell tracking and cell-based analysis

The tracking algorithm tracks individual convective cells and their volume based on the model output fields of vertical velocity and total condensate mixing ratio. The tracking of maxima in the column vertical velocity field is performed using trackpy (Allan et al., 2016). The algorithm from trackpy that is used to identify the updraft features requires an initial assumption for the size of the tracked object. We chose a diameter of $15\,\mathrm{km}$ to represent the large convective updrafts in the supercell cases.

Tracked updrafts are required to exist for six output time steps, i.e. 30 minutes, to be included in the analysis, which helps to exclude spurious features in vertical velocity and thus focus on the analysis of properly developed deep convective cells. We extrapolate by two time steps at the beginning and at the end of each tracked trajectory to include a representation of the initial development of the convective clouds and the evolution after the weakening of the central updraft.

The volume of the convective clouds is determined by a watershedding algorithm using a fixed threshold to determine the extent of the individual clouds based on the tracked updrafts. We use a threshold of $1\,\mathrm{g\,cm^{-3}}$ for the total water content in this study and a variation of this threshold by an order of magnitude to $0.1\,\mathrm{g\,cm^{-3}}$ showed that choosing a lower threshold did not significantly change the cloud volume and cloud mass or any of the more detailed process analyses.

## A2   Microphysics schemes and process rate diagnostics

Table A1 and Table A2 give an overview of the microphysical process rates for the hydrometeor masses as they are implemented in the two bulk microphysics schemes (Thompson et al., 2008; Morrison et al., 2009) studied in this paper.

In the Thompson scheme, some of the process rates are defined as signed variables representing two opposed processes. In these cases, we have used the process rate variable with the positive sign for the respective process and ignored the values with the negative sign, which are covered by the opposing process (e.g. PRG_RCG for riming of rain on graupel and PRR_RCG for

melting of graupel due to the collection by rain). Condensation/Evaporation processes and Deposition/Sublimation processes are only defined through one combined process rate variable in the code. We have thus added the process rates with a negative sign as a variable in our diagnostics (e.g. E_PRW_VCD for the evaporation of droplets in addition to PRW_VCD for condensation) to allow for independent analyses of these, e.g. when aggregating the variables in space or time.

Ice multiplication according to the Hallet-Mossop process is implemented differently in the two bulk microphysics schemes.

In the Morrison scheme, this is implemented as a direct transfer of water mass from the liquid phase to ice particles and considered as contributing to riming. In the Thompson scheme, however, it forms a transfer from the frozen hydrometeor to new ice particles and thus part of the "ice processes". Hence, these processes are found in different categories in the two tables presenting the process rates. As the actual mass transfer is negligibly small this difference between the schemes is not relevant for the analyses performed in this study.

In the Morrison microphysics scheme as used in this study, the autoconversion of cloud droplets and accretion by rain are parametrised based on Khairoutdinov and Kogan (2000) and ice nucleation is based on Cooper (1986); Rasmussen et al. (2002). The Thompson scheme applies an autoconversion parametrisation based on Berry and Reinhardt (1974), while the different freezing modes follow Bigg (1953), Cooper (1986) and Koop et al. (2000).

The two bulk microphysics schemes differ in important parameters regarding the different hydrometeor classes. The Morrison

microphysics scheme is used in its configuration that treats the dense frozen hydrometeors as hail with a density of $900\,\mathrm{kg\,m^{-3}}$, while the simulations with the Thompson microphysics used graupel with a density of $500\,\mathrm{kg\,m^{-3}}$. The density of cloud ice, however, is higher in the simulations with the Thompson scheme $890\,\mathrm{kg\,m^{-3}}$ compared to the Morrison scheme ($500\,\mathrm{kg\,m^{-3}}$), while snow density is set to $100\,\mathrm{kg\,m^{-3}}$ for both schemes. The Thompson scheme has a more complex treatment of the snow hydrometeor class compared to the Morrison scheme, making use of a combination of two size distributions and thus allowing

for a variation of the density over its evolution (Field et al., 2005; Thompson et al., 2008). The fall speed calculations are based on different equations in the two microphysics schemes, all parameters for the hydrometeor classes are left at their default values.

*Author contributions.* M.H., B.W., L.L. and P.S. designed the experiment, M.H. and B.W. implemented the microphysical pathway analysis
in WRF, M.H. set up the simulations and developed the data analysis including the tracking algorithm, M.H. wrote the manuscript and B.W., P.S. and L.L contributed to the analysis and the manuscript.

*Competing interests.* The authors declare that they have no conflict of interest.

*Code availability.* The WRF model is publicly available and can be downloaded from http://www2.mmm.ucar.edu/wrf/users/download/get_sources.html. The code of the modified WRF model with the microphysical pathway diagnostics for the two bulk microphysics schemes and
the additional second supercell case is available from the authors on request along with postprocessing code for the process rate analysis in python. The tracking algorithm applied in this study is hosted on GitHub (https://github.com/mheikenfeld/cloudtrack). The version or the tracking code used in this paper is available as  It makes use of trackpy (Allan et al., 2016), which is available on GitHub (https://github.com/soft-matter/trackpy).

*Acknowledgements.* M.H acknowledges funding from the NERC Oxford DTP in Environmental Research (NE/L002612/1). The research
leading to these results has received funding from the European Union's Seventh Framework Programme (FP7/2007-2013) project BACCHUS under grant agreement n° 603445 (M.H., P.S. and L.L). The authors acknowledge funding from the European Research Council project ACCLAIM (P.S. and B.W) under grant agreement n° 28002 from the European Union's Seventh Framework Programme (FP7/2007-2013). P.S. and M.H. acknowledge funding from the European Research Council project RECAP under the European Union's Horizon 2020 research and innovation programme with grant agreement 724602. L.L. has also been supported by CNES. B.W. has also received funding
from the Australian Government through the Australian Research Council.
The authors would like to acknowledge the use of the University of Oxford Advanced Research Computing (ARC) facility (doi:10.5281/zenodo.22558) in carrying out this work.
We want to thank Hugh Morrison and Greg Thompson for important discussions about the two bulk microphysics schemes used in the study and Matthew Christensen for helpful comments on the manuscript.

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

**Table A1.** Mass transfer process rates for the Morrison microphysics scheme (Morrison et al., 2009)

| Variable | Description | from | to | Grouping |
|----------|-------------|------|-----|----------|
| PCC | Condensation on droplets | vapour | droplets | Condensation |
| EPCC | Evaporation of droplets | droplets | vapour | Evaporation |
| PRE | Evaporation of rain | rain | vapour | |
| PRC | Autoconversion | droplets | rain | Rain formation |
| PRA | Accretion | droplets | rain | |
| MNUCCC | Contact freezing of droplets | droplets | ice | Freezing |
| MNUCCD | Primary ice nucleation | droplets | ice | |
| QICF | Homogeneous freezing of droplets | droplets | ice | |
| MNUCCR | Contact freezing of rain | rain | ice | |
| QGRF | Homogeneous freezing of rain | rain | graupel | |
| QNIRF | Homogeneous freezing of rain | rain | snow | |
| PSACWS | Riming on snow | droplets | snow | Riming |
| PSACWI | Riming on ice | droplets | ice | |
| PSACWG | Collection of droplets by graupel | droplets | graupel | |
| PGSACW | Collection of droplets by snow | droplets | graupel | |
| PRACS | Rain-snow collection | rain | snow | |
| PIACR | Ice-rain collision | rain | graupel | |
| PIACRS | Ice-rain collision | rain | snow | |
| PRACG | Collection of rain by graupel | rain | graupel | |
| PGRACS | Collection of rain by snow | rain | graupel | |
| QMULTG | Ice multiplication droplets and graupel | droplets | ice | |
| QMULTS | Ice multiplication droplets and snow | droplets | ice | |
| QMULTRG | Ice multiplication rain and graupel | rain | ice | |
| QMULTR | Ice multiplication rain and snow | rain | ice | |
| PGMLT | Melting of graupel | graupel | rain | Melting |
| QIIM | Melting of ice | ice | droplets | |
| PSMLT | Melting of snow | snow | rain | |
| PRD | Deposition on ice | vapour | ice | Deposition |
| PRDS | Deposition on snow | vapour | snow | |
| PRDG | Deposition on graupel | vapour | graupel | |
| EPRDG | Sublimation of graupel | graupel | vapour | Sublimation |
| EVPMG | Graupel melting and evaporating | graupel | vapour | |
| EPRD | Sublimation of ice | ice | vapour | |
| EPRDS | Sublimation of snow | snow | vapour | |
| EVPMS | Snow melting and evaporating | snow | vapour | |
| PRAI | Accretion of cloud ice by snow | ice | snow | Ice processes |
| PRCI | Autoconversion of cloud ice to snow | ice | snow | |
| PRACI | Ice-rain collection (ice to graupel) | ice | graupel | |
| PRACIS | Ice-rain collision (ice to snow) | ice | snow | |
| PSACR | Collection of snow by rain | snow | graupel | |

**Table A2.** Mass transfer process rates for the Thompson microphysics scheme (Thompson et al., 2008).

[*] denotes processes that are implemented but disabled in the microphysics scheme

| Variable | Description | from | to | Grouping |
|---|---|---|---|---|
| PRW_VCD | Condensation | vapour | droplets | Condensation |
| PRV_REV | Evaporation of rain | vapour | droplets | Evaporation |
| E_PRW_VCD | Evaporation of cloud droplets | droplets | vapour | |
| PRR_WAU | Autoconversion | droplets | rain | Rain formation |
| PRR_RCW | Accretion | droplets | rain | |
| PRI_WFZ | Freezing of cloud droplets | droplets | ice | Freezing |
| PRI_RFZ | Freezing of rain to ice | rain | ice | |
| PRG_RFZ | Freezing of rain to graupel | rain | graupel | |
| PRS_SCW | Collection of droplets by snow | droplets | snow | Riming |
| PRG_SCW | Collection of droplets by snow | droplets | snow | |
| PRG_GCW | Collection of droplets by graupel | droplets | graupel | |
| PRG_RCG | Collection of rain by graupel | rain | graupel | |
| PRR_RCS | Collection of rain by snow | rain | snow | |
| PRR_RCS | Collection of rain by snow | rain | graupel | |
| PRR_RCI | Collection of ice by rain | rain | graupel | |
| PRW_IMI | Melting of ice | ice | rain | Melting |
| PRR_GML | Melting of graupel | graupel | rain | |
| PRR_RCS | Collection of snow by rain | snow | rain | |
| PRR_RCG | Collection of graupel by rain | graupel | rain | |
| PRS_SDE | Deposition on snow | vapour | snow | Deposition |
| PRS_IDE | Deposition on ice to snow | vapour | snow | |
| PRI_IDE | Deposition on ice | vapour | ice | |
| PRG_GDE [*] | Deposition on graupel | vapour | graupel | |
| PRI_INU | Ice nucleation | vapour | ice | |
| PRI_IHA [*] | Freezing of aqueous aerosols | vapour | ice | |
| E_PRS_SDE | Sublimation of snow | snow | vapour | Sublimation |
| E_PRI_IDE | Sublimation of ice | ice | vapour | |
| E_PRG_GDE | Sublimation of graupel | graupel | vapour | |
| PRS_SCI | Collection of ice by snow to graupel | ice | graupel | Ice processes |
| PRS_IHM | Hallet-Mossop process | snow | ice | |
| PRS_IAU | Autoconversion of ice to snow | ice | snow | |
| E_PRS_RCS | Collection of snow by rain | snow | graupel | |
| PRI_RCI | Collection of ice by rain | ice | graupel | |
| PRG_IHM | Hallet-Mossop process | graupel | ice | |