# Peer review of "Aerosol effects on deep convection: The propagation of aerosol perturbations through convective cloud microphysics"

_Atmospheric Chemistry and Physics, 2018_

## Referee Comment (RC1) · Anonymous Referee #1 · 5 Sep 2018

This paper runs simulations of two different supercells using a suite of microphysics schemes and CCN/CDNC concentrations. They added outputs of microphysical process rates for two of the three microphysics schemes to investigate mechanisms of convective invigoration. Aerosol-induced convective invigoration is currently not well understood, and this paper contributes to the ongoing discussion in the literature on this topic. I recommend minor revisions.

Major Comments:

1. The microphysical process analysis (section 3.2) seems largely disconnected from the cloud mass and centre of gravity analysis (section 3.3). It would be nice if the microphysical process analysis could be used to help explain the results in section 3.3 more. Such a linkage also seems to be part of the goal of the paper which as stated

by the authors is "to unravel the microphysical mechanisms responsible for . . . aerosol effects on convection".

Minor Comments:

1. The authors may consider changing the title. After reading the paper I understand what is meant by the title, but I don't know that I understood it beforehand. Just a suggestion.

2. I think that the goal of the paper could be stated more clearly. It isn't explicitly stated until the conclusions that the primary aerosol effect that the authors wish to investigate is convective invigoration.

3. I don't understand how fixing the CDNC helps to "isolate" the impact of microphysical pathways. Can the authors clarify what they mean?

4. The description of the cell tracking algorithm is brief. Can the authors comment on how they handle splitting and merging of convective cells? Splitting is of particular importance to this paper given that they are simulating supercells.

5. I generally like the use of the pie charts on the cross-sections for quickly assessing the relative importance of various processes or hydrometeor amounts. That said, the authors spend a good deal of time discussing the specifics of these figures. I found myself spending a lot of time squinting at the panels, and they were difficult to use for more quantitative analysis. I'm not sure that there is a way to avoid these issues, so I just want to raise them as a comment.

6. Most of the processes in the figures are self-explanatory, but can the authors define "ice processes"?

7. Page 9, Line 1: I struggle to identify two distinct regions.

8. Page 11, Line 2: By "cloud droplets" do the authors mean number or mass?

9. Page 11, Line 4: Can the authors comment specifically on how the definitions of

hydrometeor classes differ and how these differences influence the results?

10. Page 11, Line 7: I assume that the authors track the right-mover of the supercell, but this is not stated explicitly.

11. Page 11, Lines 12-18: Try as I might, I can't see deposition anywhere on Figure 4 (or Fig. 2) so it is difficult to assess the accuracy of these statements.

12. So Figure 9 shows the results from all tracked cells? Why the switch now from looking at just one cell to all the cells?

13. Page 22, Line 15: It was very difficult to tell from the analysis as presented whether there is a near complete transfer of (liquid) condensate mass into the ice phase or not.

14. Many studies have been performed that investigated the impact of aerosols on deep convection, including some that have shown microphysical process rates. I think that generally the authors could do a better job of discussing how their results agree or disagree with these previous studies.

---

## Referee Comment (RC2) · Anonymous Referee #3 · 15 Dec 2018

Review of Heikenfeld et al., "The propagation of aerosol perturbations in convective cloud microphysics"

General comments

The authors present an analysis of microphysical processes in idealized simulations of deep convective clouds for different aerosol concentrations and three different microphysics schemes. Novel visualization techniques are presented to show the temporal and spatial evolution of the processes and the associated latent heating. A focus of the analysis is whether the "invigoration hypothesis" by Rosenfeld et al. (2008) can be confirmed (and in can not).

This last point is quite interesting and the main reason why I recommend this paper
for publication. The manuscript is very well written, and the plots are clear (though a bit small for my taste). The comparison of the microphysics schemes doesn't go into depth, and it is a bit unclear what the intention behind the presentation of three schemes is. In particular, the third scheme (SBM) is only shown for a subset of the analyses, although it deviates substantially from the other two. I recommend changes to clarify these points.

Detailed comments

- The abstract mentions that three schemes are used, but not what the benefits of the comparison are. Do they give consistent results regarding the invigoration effect? Can anything be learned from the comparison (e.g. regarding depositional growth of different ice species, which has caused a huge difference)?

- page 3, line 11-16: here the logical flow is unclear. Why is there a separate paragraph on Glassmeier and Lohmann? This needs an introductory sentence.

- The (main) text is not very clear about how many cells are simulated and how the analysis is done when there are two cells. (I assume that you have always either one or two cells, and that the properties of the two cells are averaged, but I have not found this clearly in the text. Maybe I just missed it.)

- The model setup description needs more information to make the study reproducible. In particular, Weisman and Klemp (1982, 1984) describe several versions of their idealized sound (different values of qv0), which one is used here? and how exactly is the warm bubble defined? What boundary conditions (open/fixed/periodic) are used? Such information could be given in the appendix.

- What regions/clouds are the two different model setups representative for?

- Can you comment on whether the CDNC concentrations as listed in Table 1 are actually prescribed at all grid points where there is liquid water, or only at cloud base/when new droplets form?

- Figure 2 and others: some of the pie charts are very small. Is the reader expected to read these?

- Figure 2: "contour lines for . . . ice (grey) content": Is this just cloud ice or cloud ice + snow + graupel + hail?

- Figure 2(e): It looks like there is melting above the melting level?

- Why is there no plot as Fig. 2/3/4 (and more) for the SBM scheme?

- page 11, line 31: Can you comment on which parameterizations are used for rain freezing vs. cloud drop freezing, and why one is more CCN-dependent than the other?

- Figure 10: There is a substantial difference in evaporation between the two schemes. Why is this? Mixing assumption?

- Page 18: Why is the cloud dissipating with Thompson microphysics? This is a very substantial difference that should be discussed more.

- It remains a bit unclear to me what the conclusion from the second case is. Are the result regarding the invigoration hypothesis robust? Or is everything so different that not much can be concluded from two cases and one would actually need many more?

- The conclusions could be more quantitative regarding the invigoration effect by giving number for the percentage change in latent heating.

Technical comments:

- page 1, line 24 and many other occurrences: I think it is common to list multiple references for the same statement either in chronological or in reverse chronological order, not in arbitrary order as here.

- page 5, line 6: scheme -> schemes

- page 6, caption of Table 1: "10 g/, kg-1": change "/," to the latex command "\,"

- page 14, line 8: "The differences are in part caused by . . .": This seems to be a

repetition, the same was already said in line 4.

- page 23, line 15: full stop missing after "framework".
* * *

---

## Author Comment (AC1) · 5 Feb 2019

**Authors' response to Anonymous Referee #1**

We would like to thank the reviewer for their detailed comments and suggestions on the submitted manuscript. The feedback has pointed out important aspects that required additional clarity or information and helped us a lot in improving these points in the revised manuscript.

In the following, we respond to reviewer's comments in **black**, with our answers to the comments in blue and the adapted text from the revised manuscript in green.

We have attached the revised version of the manuscript with tracked changes to the general authors' response. In the general authors' response (AR), we have added a few additional comments regarding the revised manuscript and points raised by both reviewers.

**This paper runs simulations of two different supercells using a suite of microphysics schemes and CCN/CDNC concentrations. They added outputs of microphysical process rates for two of the three microphysics schemes to investigate mechanisms of convective invigoration. Aerosol-induced convective invigoration is currently not well understood, and this paper contributes to the ongoing discussion in the literature on this topic. I recommend minor revisions.**

**Major Comments:**

**1. The microphysical process analysis (section 3.2) seems largely disconnected from the cloud mass and centre of gravity analysis (section 3.3). It would be nice if the microphysical process analysis could be used to help explain the results in section 3.3 more. Such a linkage also seems to be part of the goal of the paper which as stated by the authors is "to unravel the microphysical mechanisms responsible for aerosol effects on convection".**

We have significantly revised section 3.3 by using the findings from section 3.2 more directly in explaining some of the effects of the choice of CDNC value or microphysics scheme on the bulk cloud properties through the differences we found in the detailed process rate analysis. This provides a clearer link between the two types of analysis in the paper.

"The evolution of the cloud mass and the mass of the two water phases in the cloud (Fig. 12) in the three microphysics schemesis similar, with a maximum cloud mass of about $2 \cdot 10^{10}$ kg for all microphysics schemes before the splitting of the cell and then about $1.5 \cdot 10^{10}$ kg for the two bulk microphysics schemes (Fig. 12 a,b) and slightly higher cloud masses of up to $1.8 \cdot 10^{10}$ kg in the spectral bin microphysics scheme (Fig. 12 c). The cloud mass and also the difference between the bulk schemes and the bin scheme are dominated by the ice-phase hydrometeors, while the liquid-phase mass is very similar in all three different microphysics schemes, making up about 20-25% of the total cloud mass.
(Page 20, line 24)

There are, however, marked differences in the response to changes of the aerosol proxy between the microphysics schemes. The Morrison scheme shows a decrease of total cloud mass and ice-phase mass by about 10-15% over the range in which we increase the CDNC and no significant changes in the liquid phase. This decrease in ice-phase mass can be directly linked to the changes in the microphysical process rates analysed in Sec. 3.2. The shift of freezing to higher altitudes leads to a reduction in frozen hydrometeors in the mixed phase of the cloud and thus significantly less growth of the ice phase through vapour deposition. In the Thompson scheme, however, increased CDNC leads to an increase in ice-phase and total mass and a small increase in cloud liquid mass. This increase agrees well with the increased deposition due to the changes in the ice hydrometeor

partition in the cloud discussed in Sec. 3.2. In the simulations using the SBM scheme, the two phases show a differing response to the aerosol proxy with increased liquid hydrometeor mass and a decrease in ice-phase mass for increasing CCN. 3.2." (Page 21, line 5)

"There is a consistent response in the cloud heights for all three microphysics schemes. The microphysics schemes show an increase in the height of the centre of gravity of the entire cloud, which is more pronounced using the Thompson scheme (about 1.5 km) than in the Morrison scheme(about 0.5-1 km). This includes an upward shift in both the liquid and frozen water in the cloud. The increased height of the liquid phase can be directly related to the decrease in the formation of warm rain (Fig. 6) and the more numerous cloud droplets reaching higher up in the cloud in the polluted case compared to the dominating raindrops in the cleanest case (Fig.5). The increase in the altitude of the ice phase in the cloud with increased CDNC can be related to the changes in the altitude of the freezing processes. However, it can also be a result of the lower fall speeds of the ice and snow hydrometeors dominating in the polluted case instead of graupel and hail in the cleanest cases." (Page 22, line 2)

**Minor Comments:**

**1. The authors may consider changing the title. After reading the paper I understand what is meant by the title, but I don't know that I understood it beforehand. Just a suggestion.**

Thanks for this comment, we have decided to adapt the title to state the purpose of paper more clearly:

"Aerosol effects on deep convection: The propagation of aerosol perturbations through convective cloud microphysics"

**2. I think that the goal of the paper could be stated more clearly. It isn't explicitly stated until the conclusions that the primary aerosol effect that the authors wish to investigate is convective invigoration.**

We have adapted the respective part of the introduction to state more clearly that investigating the hypothesis of convective invigoration is one of the main focusses, but not the only focus of this study. The results show that the effects of aerosols are a complex superposition of different changes in the microphysics and the individual components of the latent heating. The integrated changes in freezing turn out to be much smaller than other changes to the latent heat release.

We have adapted the relevant sentence in the introduction of the paper to state the aim of the paper regarding the study of convective invigoration more clearly in the revised manuscript:

"It is, therefore, one of the main goals of this paper to investigate if and how these proposed mechanisms of convective invigoration, especially the proposed invigoration of convection due to additional latent heat release from freezing, manifest themselves in numerical simulations."
(Page 3, line 1)

**3. I don't understand how fixing the CDNC helps to "isolate" the impact of microphysical pathways. Can the authors clarify what they mean?**

We chose to use the fixed CDNC versions of the two microphysics schemes to exclude the extra step of cloud droplet activation in this analysis. Versions of the two microphysics schemes with activation based on prescribing the CCN exist, however, these are implemented in a different way in the two schemes, which would add additional differences between the microphysics schemes.

We have worded that more clearly in the respective paragraphs in the introduction and the methodology:

"To isolate the role of cloud microphysics for aerosol effects on deep convection from additional uncertainties in model-simulated aerosol fields, we apply a fixed cloud droplet number concentration (CDNC) in the two bulk microphysics schemes for each simulation. In each of the schemes, the CDNC is reset to the chosen value at the end of each model time step in all cloudy grid points. We vary this CDNC value between different simulations as a proxy for aerosol number concentration. There are versions of both bulk microphysics schemes that include the activation of a fixed CCN spectrum or even interactive aerosols (Thompson and Eidhammer, 2014; Wang et al., 2013). However, the implementation of both the cloud droplet activation and the representation of the aerosol distributions is very different between the two microphysics schemes, which would add additional differences between the schemes compared to representing the perturbations in the form of a varying CDNC." (Page 5, line 14)

**4. The description of the cell tracking algorithm is brief. Can the authors comment on how they handle splitting and merging of convective cells? Splitting is of particular importance to this paper given that they are simulating supercells.**

Cell splitting and merging is not explicitly treated in the tracking algorithm we are using here. The tracking algorithm picked up the initial updraft in the cell before splitting and then followed the right-moving cell after the split, while the updraft of the left moving got picked up as a new feature. For this study, this is not an issue as we entirely focus on the microphysical evolution of one of the cells. The way the cell splits seems to be very similar between the different simulations and not strongly affected by the choice of the microphysics scheme or the chosen value for the CDNC/CCN, so we did not analyse this further. The second cell moving to the left of the initial cell direction is not analysed in detail here as it shows very similar results in all aspects discussed. See also answer to comment 3 by Referee #3.

We have extended the description of the tracking algorithm and our choice of cell in the analysis in the revised manuscript to make this clearer to the reader:

"The tracking algorithm does not explicitly treat splitting and merging of convective cells. In all simulated cases in this study, the initial convective cell splits into two separate counter rotating cells early into the simulations. In CASE1 this leads to a relatively symmetric situation with similarly strong individual cells. In both cases, one of the cells develops more directly out of the initial cell, in CASE1 this is the right-moving cell, while in CASE2 this is the stronger left moving cell. In each simulation, this stronger cell gets picked up as a continuation of the initial cell by the tracking algorithm. The second cell has been analysed following the same methodology and showed very similar results in all major aspects. We have thus decided to focus on the analysis of the first cell in this paper and to not discuss the results from the second cell in more detail." (Page 9, line 12)

**5. I generally like the use of the pie charts on the cross-sections for quickly assessing the relative importance of various processes or hydrometeor amounts. That said, the authors spend a good deal of time discussing the specifics of these figures. I found myself spending a lot of time squinting at the panels, and they were difficult to use for more quantitative analysis. I'm not sure that there is a way to avoid these issues, so I just want to raise them as a comment.**

We thank the reviewer for raising this important point that we have also thought about quite a lot when developing the analyses for this paper. We agree that there is a price to pay in the trade-off between a straightforward quantitative analysis and getting the full picture that the two-dimensional presentation with pie charts gives for assessing the structure and time evolution in a vertically resolved way. We have significantly increased the size of many of the pie charts in the revised

manuscript by increasing the figure sizes or reducing the axis ranges, which makes the figures much easier to read.

**6. Most of the processes in the figures are self-explanatory, but can the authors define "ice processes"?**

The processes grouped as "Ice processes" combine all processes transferring mass between the different frozen hydrometeors (e.g. autoconversion of ice particles, collection of cloud ice by snow, etc., see also Table A1 and A2 in the appendix).

We have included a paragraph explaining the grouping of the individual processes depicted in these figures in the revised manuscript:

"For most analyses in this study, the individual microphysical processes are grouped into a consistent set of classes according to their contribution to the hydrometeor mass transfer in the model. This includes the six different phase transitions between frozen hydrometeors, water drops and water vapour (*condensation, evaporation, freezing including riming, melting, deposition* and *sublimation*) as well as the warm *rain formation* due to autoconversion and accretion of cloud droplets and all processes that transfer mass between the different frozen hydrometeors as *ice processes*. For some of the more detailed analyses, this grouping is performed in a more detailed way, e.g. separating freezing and riming processes or splitting them up by the specific hydrometeor class involved in the transfer. A collection of all the individual microphysical process rates represented in the two bulk microphysics schemes including the grouping discussed here is given in the appendix (Table A1 for the Morrison microphysics scheme and in Table A2 for the Thompson microphysics scheme)."
(Page 7, Line 10)

**7. Page 9, Line 1: I struggle to identify two distinct regions.**

We agree, that "distinct regions" is probably a bit overstated. Still, there is a significantly larger vertical range over which freezing and riming occur in the Morrison scheme, with maxima around these two heights (also visible in the time evolution for the clean case in Fig. 6). We rephrased the text in the revised manuscript:

"During the later stage, the freezing in the simulation using the Morrison microphysics scheme takes place over a substantial vertical range and is strongest at both edges of the mixed-phase region of the cloud at around 8 km and 10 km altitude (Fig. 2 c)." (Page 11, line 4)

**8. Page 11, Line 2: By "cloud droplets" do the authors mean number or mass?**

We mean cloud droplet mass, we have adapted that accordingly in the text. (Page .., line ..)

**9. Page 11, Line 4: Can the authors comment specifically on how the definitions of hydrometeor classes differ and how these differences influence the results?**

We have added additional information on the specification of the hydrometeor classes in the introduction and provided more details about it at the relevant parts in the discussion.

One important point is the difference in the parametrisation of individual microphysical processes, or even the existence of processes as in the case of deposition on graupel in the Thompson scheme. We have addressed the impact of that difference more detailed through the inclusion of an additional figure (Fig. 10) and extended discussions of the implications for specific microphysical processes.

"In the simulations with the Thompson microphysics scheme (Fig. 9 c,d), deposition and sublimation processes show very a different behaviour. The strong increase in snow in the cloud with increasing CDNC (Fig. 5 c,d) leads to a strong increase in both deposition and sublimation on

snow. Deposition on ice is on the same order of magnitude for the cleanest case, but not strongly affected by a change in CDNC. Sublimation of graupel only occurs around and below the melting layer and is significantly reduced by increasing CDNC. As deposition on graupel is prohibited in this microphysics scheme, there is no decrease in deposition on graupel associated with the changes in the hydrometeor ratio compensating the increase in deposition on snow. This leads to a strong increase in total deposition with increased CDNC as the main response in the Thompson scheme."
(Page 16 , line 13)

We have added additional details on the definition of the hydrometeor classes and important differences between the bulk schemes in the appendix describing the microphysics schemes in more detail:

"The two bulk microphysics schemes furthermore differ in important parameters regarding the different hydrometeor classes. The Morrison microphysics scheme is used in its configuration that treats the dense frozen hydrometeors as hail with a density of 900 kg m$^{-3}$ , while the simulations with the Thompson microphysics used graupel with a density of 500 kg m$^{-3}$ . The density of cloud ice, however, is higher in the simulations with the Thompson scheme 890 kg m$^{-3}$ compared to the Morrison scheme (500 kg m$^{-3}$ ), while snow density is set to 100 kg m $^{-3}$ for both schemes. The Thompson scheme has a more complex treatment of the snow hydrometeor class compared to the Morrison scheme, making use of a combination of two size distributions and thus allowing for a variation of the density over its evolution (Field et al., 2005; Thompson et al., 2008). The fall speed calculations are based on different equations in the two microphysics schemes, all parameters for the hydrometeor classes are left at their default values."
(Page 30, line 129)

 We have added more details on the role of the representation of hydrometeors based on distinct classes and the resulting challenges in the introduction of the paper:

"The separation of the hydrometeors into individual hydrometeor classes in microphysics schemes brings with it specific challenges in resolving the microphysical processes. In bulk schemes, liquid water in the cloud is separated into cloud droplets and raindrops. The collision-coalescence processes leading to the formation of rain from cloud droplets have to be parametrised through the artificial process of droplet autoconversion and a simplified treatment of accretion of droplets by raindrops. The semi-empirical nature of these parametrisations has been shown to be the source of major uncertainty in the assessment of aerosol-cloud interactions in numerical model simulations (Khain et al., 2015; White et al., 2017). In the ice phase, most current microphysics schemes separate the hydrometeors into a number of different classes such as pristine ice, snow, hail or graupel. The equations and parameters for the calculation of the microphysical process rates as well as important physical properties of the hydrometeors, such as shape, density or the specific form of the size distribution are specified for each individual hydrometeor class. These choices additionally impact important physical processes such as the fall speeds of hydrometeors in the calculation of sedimentation or the radiative properties of the hydrometeors. This can lead to abrupt changes to the evolution of the cloud due to a change in the partition between the hydrometeor classes in the ice phase of the cloud (Morrison and Milbrandt, 2014)" (Page 3, line 14)

**10. Page 11, Line 7: I assume that the authors track the right-mover of the supercell, but this is not stated explicitly.**

We track both cells, but we have only analysed the right-moving cell including the initial stage. We have added a clearer description of the tracking and analysis in the revised manuscript (see also comment 4 for more details).

We have amended the text here and at some other points to state this more clearly:

"As for all the following figures for CASE1, these analyses are based on a combination of the initial stage of the cell and the right-moving cell after the cell split." (Page 9, line 27)

**11. Page 11, Lines 12-18: Try as I might, I can't see deposition anywhere on Figure 4 (or Fig. 2) so it is difficult to assess the accuracy of these statements.**

The enlarged figures and choice of colours makes it easier to distinguish the individual process rates. The deposition processes should now be clearly visible, especially in the panels showing the latent heat release from the processes (e.g. Fig 2 f on page 10).

**12. So Figure 9 shows the results from all tracked cells? Why the switch now from looking at just one cell to all the cells?**

Throughout the entire paper, we only analyse one of the two tracked cells (see response to comments 4 and 10). We acknowledge that the use of the plural "cells" for the cell in the different cases/microphysics schemes might be misleading, so we have adapted this in the revised manuscript and clarified the respective figure captions (Fig. 12, Fig. 15):

"Total water mass, liquid water mass and frozen water mass in the analysed right-moving cell for the three different microphysics schemes (Morrison: left, Thompson: middle, SBM: right) in CASE1. The jump in the curves occurs at the point where the cell splits into two individual cells"
(Page 22, caption Fig.12 )

**13. Page 22, Line 15: It was very difficult to tell from the analysis as presented whether there is a near complete transfer of (liquid) condensate mass into the ice phase or not.**

This statement was based on the fact that the cloud hydrometeor mass is predominantly made up of ice-phase hydrometeors (Fig.11 and Fig. 14). However, the significant changes in the formation of rain from cloud droplets observed in all microphysics schemes show that there is a significant contribution of warm rain processes to precipitation. Reducing the precipitation indeed gives a significant potential for the invigoration pathway to occur (through additional freezing), whatever the partition between liquid and frozen water in the cloud. We have thus removed this statement from the revised manuscript.

**14. Many studies have been performed that investigated the impact of aerosols on deep convection, including some that have shown microphysical process rates. I think that generally the authors could do a better job of discussing how their results agree or disagree with these previous studies.**

We have added additional discussion of the results in light of previous studies of aerosol effects on supercells and other isolated deep convective clouds in the conclusions section of the paper, e.g. in the following sections:

"This response is consistent between the different microphysics schemes and confirms earlier studies that stated the importance of changes in the partition between rain and cloud droplets in determining the evolution of freezing and riming (Seifert and Beheng, 2006)."
(Page 27, line 25)

"This confirms results from previous studies on the effects of aerosols on supercells (Khain et al., 2008; Morrison, 2012; Kalina et al., 2014) and other deep convective clouds (Ekman et al., 2011) that pointed out a range of compensating processes limiting convective invigoration and a strong dependency on the environmental conditions in which the cloud develops."
(Page 28, line 1)

---

## Author Comment (AC2) · 5 Feb 2019

**Authors' response to Anonymous Referee #3**

We would like to thank the reviewer for their detailed comments and suggestions. The feedback pointed out important aspects that required additional clarity or information and helped us a lot to improve these points in the revised manuscript.

In the following, we respond to the reviewer's comments in **black**, with our answers to the comments in blue and the adapted text from the revised manuscript in green.

We have attached the revised version of the manuscript with tracked changes to the general authors' response. In the general authors' response (AR), we have added a few additional comments regarding the revised manuscript and points raised by both reviewers.

**General comments:**

**The authors present an analysis of microphysical processes in idealized simulations of deep convective clouds for different aerosol concentrations and three different microphysics schemes. Novel visualization techniques are presented to show the temporal and spatial evolution of the processes and the associated latent heating. A focus of the analysis is whether the "invigoration hypothesis" by Rosenfeld et al. (2008) can be confirmed (and in can not).**
**This last point is quite interesting and the main reason why I recommend this paper for publication. The manuscript is very well written, and the plots are clear (though a bit small for my taste).**
**The comparison of the microphysics schemes doesn't go into depth, and it is a bit unclear what the intention behind the presentation of three schemes is. In particular, the third scheme (SBM) is only shown for a subset of the analyses, although it deviates substantially from the other two. I recommend changes to clarify these points.**

We answer to the points raised here (size of the pie chart plots and the choice of analysis for the three microphysics schemes) in more detail where they were raised in the respective detailed comments.

**Detailed comments:**

**1. The abstract mentions that three schemes are used, but not what the benefits of the comparison are. Do they give consistent results regarding the invigoration effect? Can anything be learned from the comparison (e.g. regarding depositional growth of different ice species, which has caused a huge difference)?**

We have adapted the abstract to give a clearer overview of our approach and the most important results of the analysis.

**2. page 3, line 11-16: here the logical flow is unclear. Why is there a separate paragraph on Glassmeier and Lohmann? This needs an introductory sentence.**

We have included this study in the overview of the existing literature since it provides an different approach to understanding the pathways by focussing on an analytical analysis of the equations implemented in a microphysics scheme. We have shortened this section in the revised manuscript and merged it into one paragraph with the overview of other existing studies using numerical simulations with cloud-resolving models:

"In addition to the analysis of process rates in numerical simulations, analytical evaluations of the microphysical rate equations of the microphysics schemes can give important insights into the propagation of aerosol effects in the cloud microphysics (Glassmeier and Lohmann, 2016). This
kind of analytical approach works well for warm-phase clouds but is less conclusive for the response of mixed-phase clouds, especially deep convective clouds, due to many compensating effects and the complexity of the processes involving ice-phase hydrometeors (Glassmeier and Lohmann, 2016)." (Page 2, line 14)

**3. The (main) text is not very clear about how many cells are simulated and how the analysis is done when there are two cells. (I assume that you have always either one or two cells, and that the properties of the two cells are averaged, but I have not found this clearly in the text. Maybe I just missed it.)**

The tracking algorithm identifies the updraft in the initial cell and then after the split, follows the right-moving cell for the rest of the evolution (red in Fig. 1). All our analysis follows the evolution of this combination of the initial cell and the right moving cell. The second cell (yellow in Fig.1) after the split moving leftwards is picked up as a separate cell. We performed the same analyses for that second cell (not shown) which gave very similar results. Similarly, the dominant cell in the second case, which shows a stronger asymmetry in the magnitude of the two individual cells, is used for all analyses in CASE2. See also answer to comment 4 by Referee #1.

We have adapted the text in the methods section of the revised manuscript (section 2) to explain this more clearly:

"The tracking algorithm does not explicitly treat splitting and merging of convective cells. In all simulated cases in this study, the initial convective cell splits into two separate counter rotating cells early into the simulations. In CASE1 this leads to a relatively symmetric situation with similarly strong individual cells. In both cases, one of the cells develops more directly out of the initial cell, in CASE1 this is the right-moving cell, while in CASE2 this is the stronger left moving cell. In each simulation, this stronger cell gets picked up as a continuation of the initial cell by the tracking algorithm. The second cell has been analysed following the same methodology and showed very similar results in all major aspects. We have thus decided to focus on the analysis of the first cell in this paper and to not discuss the results from the second cell in more detail." (Page 9, line 12)

**4. The model setup description needs more information to make the study reproducible. In particular, Weisman and Klemp (1982, 1984) describe several versions of their idealized sound (different values of qv0), which one is used here? and how exactly is the warm bubble defined? What boundary conditions (open/fixed/periodic) are used? Such information could be given in the appendix.**

We have revised the manuscript by adding additional information regarding the two idealised setups to the description of the modelling setup, including more detailed information about the profile and the methods used for the initiation of convection and boundary conditions:

"We simulate two different idealised supercell cases. The first set of simulations (CASE1) is based on the default WRF quarter-circle shear supercell case (Khain and Lynn, 2009; Lebo and Seinfeld, 2011) representative of a supercell case over the Southern Great Plains of the United States. This case uses an initial sounding described in Weisman and Klemp (1982) with a surface temperature of 300 K and a surface vapour mixing ratio of 14 g kg$^{-1}$ . The wind profile is taken from Weisman and Rotunno (2000) and features a wind shear of 40 m s$^{-1}$ made up of a quarter-circle shear up to a height of 2 km and a linear shear further up to 7 km height. The initiation of convection is triggered by a warm bubble with a magnitude of 3 K in potential temperature centred at 1.5 km height in the centre of the domain with a radius of 10 km horizontally and 1.5 km vertically in which the perturbation decays with the square of the cosine towards the edge of the bubble (Morrison, 2012). This type of setup has been used for a number of similar studies in the past (Storer et al., 2010; Morrison and Milbrandt, 2010; Morrison, 2012; Kalina et al., 2014).

To test the representativeness of the results for different cases of idealised deep convection, a set of simulations for a second supercell case (CASE2) is based on an observed supercell storm over Oklahoma in 2008 (Kumjian et al., 2010). In contrast to the first case, the profiles in this case are from observation used in the model experiments in Dawson et al. (2013). This case features a significantly drier initial profile with a surface temperature of 308 K and a surface water vapour mixing ratio of 16 g kg$^{-1}$ along with wind shear of similar magnitude to CASE1. The initiation of convection in this case is created by forced convergence near the surface based on nudging for the vertical velocity over the same volume that is used for the warm bubble in CASE1 according to the methodology described in Naylor and Gilmore (2012) with an updraft speed peaking at 5 m s$^{-1}$ at the centre.

Both cases are simulated without a boundary layer scheme and without the calculation of surface fluxes or radiation. The horizontal grid spacing of the simulations is 1 km to sufficiently resolve the main features of the simulated supercell. We use a model domain size of 84 grid cells in each horizontal dimension and open boundary conditions on each side of the modelling domain. The vertical resolution of the 96 model layers varies from about 50 m at the surface to 300 m at the top of the model. Simulations are performed with a time-step of 5 seconds. The standard model diagnostics and the microphysical pathway diagnostics (Section 2.3) are output every 5 minutes to sufficiently resolve the development of the microphysical processes during the life cycle of the deep convective clouds. (Page 6, line 2)

*5*. **What regions/clouds are the two different model setups representative for?**

Both cases are representative for the supercell storms over the Southern Great Plains of the US, we have added additional information on the cases to the description of the model setup. (See response to the previous comment)

**6. Can you comment on whether the CDNC concentrations as listed in Table 1 are actually prescribed at all grid points where there is liquid water, or only at cloud base/when new droplets form?**

The CDNC is prescribed everywhere in the column where there is liquid water, not only at cloud base. We have amended the text to state that more clearly.

"In each of the schemes, the CDNC is reset to the chosen value at the end of each model time step in all cloudy grid points."  (Page 5, line16)

**7. Figure 2 and others: some of the pie charts are very small. Is the reader expected to read these?**

We agree that some pie charts were too small in the initial version of the paper, thanks for pointing that out.

We have adapted most of the figures containing pie charts in terms of vertical size and the axis ranges to increase the size of the pie charts where possible. Along with the improved choice of colours (see comment 9) this strongly increases the readability of the pie charts in the revised manuscript. We have made sure we only draw conclusions from pie charts that are big enough to read them from a printed version of the paper or without zooming into a digital version of a manuscript.

It is unavoidable that some of the pie charts get small for some regions of the cloud when sticking to a representative linear relationship between coloured area and mass transfer or latent heating in the plots, as opposed to e.g. a logarithmic representation that we also tested.
Making the figures much larger would have made it difficult to place plots next to each other where different aspects of them can be compared directly, e.g. with regard to the vertical position of the microphysical processes for the different cases. However, as the size of the pie charts is representative of the total process rates, very small pie charts are indicative of regions less relevant in terms of the water turnover in the processes.

**8. Figure 2: "contour lines for . . . ice (grey) content": Is this just cloud ice or cloud ice + snow + graupel + hail?**

This contour line includes the mixing ratio of all frozen hydrometeors, we have changed the notation to "frozen (grey) water content" (Page 10, caption Fig. 2) to make it clear what we mean here.

**9. Figure 2(e): It looks like there is melting above the melting level?**

These pie charts in the centre of the cloud actually show a combination of evaporation and sublimation, but we agree that the combination really looks like the orange we chose for the melting processes.

We have adapted the choice of colours for the melting, evaporation and sublimation processes to make them more distinct and more discernible, especially when they occur in combination.
Together with the increase of the size of the pie charts in the revised manuscript (see also response to comment 7), the respective figures are much easier to read now.

**10. Why is there no plot as Fig. 2/3/4 (and more) for the SBM scheme?**

The main focus of this paper is on the understanding of the evolution of the microphysical process rates in the two bulk microphysics schemes and the impact of changes in the aerosol proxy. The spectral bin microphysics scheme has been added to set these results into the context of a third microphysics scheme with a decidedly different approach to the representation of specific processes and properties.
We have only implemented the detailed microphysical process analysis for the two bulk microphysics schemes, where these processes are explicitly described as individual process rates in the model microphysics.
A similar comparison including the same visualisation of the detailed process rates in a bin scheme would be very interesting but is beyond the scope of this study and would require substantial additional work to add the respective output to the version of the bin microphysics scheme in WRF. A direct comparison of the process rates between the bulk schemes and the bin scheme would also involve the development of a consistent mapping of the bin-resolving process rates in the bin scheme to the bulk process rates in the bulk schemes – which is far from trivial.

We have phased our approach regarding the two bulk microphysics schemes and the bin microphysics scheme more clearly in the introduction and methodology description of the revised manuscript.

"We compare the results to simulations performed with a bin microphysics scheme (HUJI spectral bin scheme) for a subset of the analyses to investigate whether the effects investigated in more detail through the microphysical pathway analysis for the two bulk microphysics schemes agree with the response of a bin microphysics scheme to perturbations of aerosol proxies."
(Page 5, line 7)

"The detailed analysis of the process rates in this paper are carried out for simulations with these two bulk microphysics schemes. To investigate how the results obtained from the detailed analysis of the two bulk microphysics schemes hold for a bin cloud microphysics scheme, we also include additional simulations with the Hebrew University cloud model (HUCM) spectral-bin microphysics scheme (Khain et al., 2004; Lynn et al., 2005a, b), called SBM in the rest of the paper. We perform a subset of the analyses for this microphysical scheme, excluding the detailed microphysical process rate analysis but including the analysis of changes to the hydrometeor mixing ratios and the bulk cloud properties."
(Page 5, line 23)

**11. page 11, line 31: Can you comment on which parameterizations are used for rain freezing vs. cloud drop freezing, and why one is more CCN-dependent than the other?**

The freezing parametrisations are given in the appendix A2. However, both freezing parametrisations do not have any dependence on droplet/drop number concentration through the effective radius. Instead, the shift from rain freezing to droplets freezing is purely related to the change in the mixing ratio of the two liquid hydrometeors with a change in CDNC.

We have stated this aspect more clearly in the revised manuscript:

"For both bulk microphysics schemes, freezing of raindrops and cloud droplets occur in two separate layers, with freezing of raindrops at around 8 km and freezing of cloud droplets above a height of 10 km up to 14 km. In both microphysics schemes, freezing of raindrops is strongly decreased for increased CDNC (Fig. 8 b,d), while freezing of cloud droplets is increased by about a factor of three. This is not related to the parametrisation of the freezing processes (described in more detail in appendix A2), which does not include any information about cloud droplet effective radius and raindrop effective radius through the number concentrations. Instead, these changes are purely a result of the shift in the abundance of cloud droplets and raindrops (Fig. 5)."
(Page 14, line 6)

**12. Figure 10: There is a substantial difference in evaporation between the two schemes. Why is this? Mixing assumption?**

The difference in evaporation between the two bulk schemes can be separated into two different components. First, the evaporation of rain at the bottom of the cloud, which decreases more strongly in the Thompson scheme due to the stronger decrease in precipitation. Second, the changes to evaporation in the higher layers of cloud from the evaporation of cloud droplets. Due to the use of saturation adjustment, the evaporation is not directly controlled by the CDNC and effective radius of the cloud droplets. However, there are strong differences in the deposition rate on frozen hydrometeors, both between the two microphysics schemes and for different CDNC values, especially in the Thompson scheme. These changes in deposition could directly affect the evaporation by significantly changing the water subsaturation in the mixed-phase region of the cloud by further reducing the water vapour in the parts of the clouds that are subsaturated with regards to water but not to ice. This is a manifestation of the Wegener-Bergeron- Findeisen process transferring water from the liquid-phase hydrometeors to the ice-phase hydrometeors.

We have amended the text in the respective paragraph to discuss the differences and changes in evaporation more clearly and further elaborate on this relationship between the evaporation and deposition processes:

"The same limitation applies to the evaporation of cloud droplets, which also cannot show any direct effect from changes in CDNC due to the use of saturation adjustment. However, the evaporation shows much stronger differences between the two microphysics schemes and also a stronger effect of a variation in CDNC (Fig. 11 b,h). The strong changes in the evaporation at higher levels in the mixed-phase region of the cloud, especially for the Thompson scheme, can be explained with the changes in deposition on frozen hydrometeors (Fig. 11 e,k). The increased deposition with increasing CDNC through the changes to the frozen hydrometeors could lead to a further decrease of the saturation vapour pressure over water in the water-subsaturated regions of the cloud and thus additional evaporation. There is also a noticeable decrease in condensation in the higher layers of the mixed-phase region of the cloud at around 10 km for the Thompson scheme (Fig. 11 g), which could be similarly related to the increase in deposition. The evaporation in the lower layers is associated with the evaporation of raindrops. The differences between the two schemes and the variation with changes in CDNC can be directly related to the differences in the amount of rain, which is both higher and more strongly decreasing with increasing CDNC in the Thompson scheme than in the Morrison scheme."
(Page 16, line 29)

"There are large differences between the microphysics schemes in the latent heating and cooling from sublimation and deposition and its response to changes in CDNC. The Morrison scheme shows a significant decrease of both sublimation and deposition with increased CDNC (Fig. 11 e,f). Apart from changes due to the shift in hydrometeors from hail to snow and cloud ice (Fig. 5 and Fig.9), these decreases can be related to the lower amount of ice hydrometeors in the mixed phase region of the cloud. Although these two changes cancel each other to a large extent in the integrated latent heating, the two processes occur at different heights, which results in a shift of latent heating to lower levels, opposing the changes to the freezing and riming processes (Fig. 11 c). Furthermore, this strong decrease in sublimation leads to a decrease in water vapour near the cloud base, which could cause the consistent decrease in condensation at around 5 km altitude in the Morrison scheme (Fig. 11 a).

In the Thompson scheme, sublimation of ice hydrometeors is weak and barely affected by changes in CDNC (Fig. 11 l). However, increases in CDNC lead to an increase in deposition in the higher parts of the cloud (Fig. 11 k). This effect can be explained by the observed shift in hydrometeors from graupel to cloud ice and snow since deposition on graupel is turned off in the Thompson microphysics scheme, while it occurs on both snow and cloud ice. This increase in deposition could be the main reason for the changes observed in evaporation of cloud droplets as it significantly increases the sub-saturation over water in the mixed phase in regions that are supersaturated with respect to ice. This can be interpreted as a manifestation of the Wegener-Bergeron-Findeisen process (Wegener, 1911; Findeisen, 1938; Findeisen et al., 2015; Storelvmo and Tan, 2015), transferring water mass from liquid hydrometeors to the frozen hydrometeors. This constitutes an additional feedback from the changes in the ice phase back onto the liquid phase hydrometeors" (Page 17, line 10)

**13. Page 18: Why is the cloud dissipating with Thompson microphysics? This is a very substantial difference that should be discussed more.**

Although we cannot rule out other dynamical explanations for this behaviour, the Thompson scheme shows much stronger cooling from the evaporation of raindrops and melting of frozen hydrometeors below cloud base, which could inhibit the later stages of the cell. This agrees with a short lifetime of the clean simulations for CASE1 with the Thompson scheme, that also show strong evaporation and melting at cloud base. We included this discussion in the revised manuscript:

"As a result, evaporation in the lowest model levels decreases strongly for the high CDNC value in the simulations with the Thompson scheme. Both microphysics schemes show a significant decrease in the total amount of melting of frozen hydrometeors below the melting line at about 4 km height. The strong cooling due to evaporation and melting in the cleanest cases for the simulations with the Thompson scheme (Fig. 6 c) can explain the significantly shorter lifetime of the cell compared to the more polluted cases and the other bulk scheme." (Page 13, line 18)

"For the Thompson microphysics scheme, this second episode of development in the tracked cell is completely absent for all simulations, with the cloud dissipating after about 60 minutes of simulation time. This is potentially related to the substantially higher cooling at and below cloud base due to the evaporation of rain and the melting of frozen hydrometeors. The cooling can substantially weaken the convective updraft and thus prevent the further development of the cell that takes place in the simulations using the two other microphysics schemes. This finding agrees with a substantially shorter lifetime of

the cleanest case for the simulations with the Thompson scheme in CASE1 (Fig. 6)."
(Page 24, Line 1)

**14. It remains a bit unclear to me what the conclusion from the second case is. Are the result regarding the invigoration hypothesis robust? Or is everything so different that not much can be concluded from two cases and one would actually need many more?**

Although the two cases are quite different, e.g. regarding the point raised in the last comment, the response of the individual microphysical processes to changes in CDNC are very similar to the ones observed in the first case. However, previous studies (e.g. Khain, 2009) have shown the wide range of responses in deep convective clouds, especially for the simulation of supercell cases.

**15. The conclusions could be more quantitative regarding the invigoration effect by giving number for the percentage change in latent heating.**

We have calculated the relative change in total latent heating with increasing CDNC and it is negligibly small (a few percent) in all simulations, there is no trend with changes in CDNC that goes beyond the small random variation the between different simulations with each microphysics scheme. This holds for both microphysics scheme and the bin microphysics scheme. The changes to individual components such as deposition or sublimation are much stronger accumulating to relative changes of up to 30 percent, which however cancel out to give no significant response in the total latent heating. The latent heat release of freezing shows no significant changes of integrated heating with CDNC, just like the total latent heating. We have added the integrated latent heating rates to Fig. 10 (Fig. 9 in the old manuscript) and discussed them in more detail in the revised manuscript:

"The changes to the vertically integrated latent heating in the cloud for all three microphysics schemes do not show a significant trend with increasing CDNC (Fig. 10 d,e,f). The Thompson scheme shows lightly higher integrated latent heating for the two simulations with the highest CDNC content, but no consistent trend over the rest of the simulations (Fig. 10 e). The SBM simulations show a slightly decreasing trend of integrated latent heating for the highest CDNC values above 1000 cm $^{-3}$ but no consistent trend over the entire range of values (Fig. 10 f) . Despite the significant change to the altitude of freezing there is no systematic change in the integrated latent heat release from freezing for both bulk microphysics schemes that would contribute to an invigoration of the cloud. In the Morrison scheme, the strong changes in deposition and sublimation almost entirely cancel out when integrated vertically. In the Thompson microphysics scheme, the increase in the integrated latent heat release from deposition cancels out the significant decrease in the integrated evaporation of cloud droplets and rain."
(Page 19, line 7)

***Technical comments:***

**1. page 1, line 24 and many other occurrences: I think it is common to list multiple references for the same statement either in chronological or in reverse chronological order, not in arbitrary order as here.**

We have revised the manuscript to order references in the same statement chronologically. Thanks for picking up this mistake.

**2. page 5, line 6: scheme -> schemes**

Corrected.

**3. page 6, caption of Table 1: "10 g/, kg-1": change "/," to the latex command "\,"**

Corrected.

*4. page 14, line 8: "The differences are in part caused by . . .": This seems to be a repetition, the same was already said in line 4.*

We have completely rephrased the paragraph which removes the repetition (See response to comment 12).

*5. page 23, line 15: full stop missing after "framework".*

Corrected.